# REST/NRSF drives homeostatic plasticity of inhibitory synapses in a target-dependent fashion

Cosimo Prestigio[1,2†], Daniele Ferrante[1,2†], Antonella Marte[1], Alessandra Romei[2], Gabriele Lignani[3], Franco Onofri[1,4], Pierluigi Valente[1,4], Fabio Benfenati[2,4*], Pietro Baldelli[1,4*]

[1]Department of Experimental Medicine, University of Genova, Genova, Italy; [2]Center for Synaptic Neuroscience and Technology, Istituto Italiano di Tecnologia, Genova, Italy; [3]Department of Clinical and Experimental Epilepsy, UCL Queen Square Institute of Neurology, Queen Square House, London, United Kingdom; [4]IRCCS, Ospedale Policlinico San Martino, Genova, Italy

*For correspondence:
benfenati.fabio@iit.it (FB);
pietro.baldelli@unige.it (PB)

†These authors contributed equally to this work

**Abstract** The repressor-element 1-silencing transcription/neuron-restrictive silencer factor (REST/NRSF) controls hundreds of neuron-specific genes. We showed that REST/NRSF downregulates glutamatergic transmission in response to hyperactivity, thus contributing to neuronal homeostasis. However, whether GABAergic transmission is also implicated in the homeostatic action of REST/NRSF is unknown. Here, we show that hyperactivity-induced REST/NRSF activation, triggers a homeostatic rearrangement of GABAergic inhibition, with increased frequency of miniature inhibitory postsynaptic currents (IPSCs) and amplitude of evoked IPSCs in mouse cultured hippocampal neurons. Notably, this effect is limited to inhibitory-onto-excitatory neuron synapses, whose density increases at somatic level and decreases in dendritic regions, demonstrating a complex target- and area-selectivity. The upscaling of perisomatic inhibition was occluded by TrkB receptor inhibition and resulted from a coordinated and sequential activation of the *Npas4* and *Bdnf* gene programs. On the opposite, the downscaling of dendritic inhibition was REST-dependent, but BDNF-independent. The findings highlight the central role of REST/NRSF in the complex transcriptional responses aimed at rescuing physiological levels of network activity in front of the ever-changing environment.

## Editor's evaluation

This manuscript investigates the role of the repressor-element 1-silencing transcription/neuron-restrictive silencer factor (REST/NRSF) in homeostatic regulation of neuronal activity in response to chronic hyperactivity in cultured neural networks. This work demonstrates the role of REST in network firing properties and specifically the role of inhibition in this regulation. This paper is of potential interest to a broad audience of neuroscientists, interested in the mechanisms of homeostatic plasticity and in brain disorders associated with neural dyshomeostasis.

## Introduction

The brain is characterized by a constant and fine homeostatic regulation of neuronal intrinsic excitability and synaptic strength aimed at keeping neuronal networks' activity within a physiological range. The dysregulation of such homeostatic mechanisms is implicated in the early phase progression of several neurological disorders, including epilepsy and Alzheimer's disease (*Frere and Slutsky, 2018*; *Lignani et al., 2020*; *Styr and Slutsky, 2018*). An increasing amount of experimental evidence shows

that the repressor element 1-silencing transcription factor (REST; also known as neuron-restrictive silencer factor, NRSF) is the molecular hub of a complex neuronal transcriptomic remodeling aimed at maintaining brain homeostasis (*Zullo et al., 2019*; *Lu et al., 2014*; *Hu et al., 2011*; *Pozzi et al., 2013*; *Pecoraro-Bisogni et al., 2018*).

REST, initially described as a nuclear negative regulator, is a zinc-finger transcription factor with several unique properties. Highly expressed in embryonic stem-cells, it is rapidly downregulated in neural progenitors and maintained at low levels after differentiation, thus enabling neurons to express the critical genes necessary for the acquisition and preservation of the neuronal phenotype (*Ballas et al., 2001*; *Ballas et al., 2005*; *Chong et al., 1995*; *Ooi and Wood, 2007*; *Schoenherr and Anderson, 1995*). REST recruits multiple corepressors, including histone deacetylases and methyl-transferases, to conserved 21 bp REST binding sites, known as RE-1 sites, on target gene promoters and represses their expression in non-neuronal cells (*Ballas et al., 2005*; *Ballas et al., 2001*; *Griffith et al., 2001*). REST targets hundreds of neuron-specific genes, including those encoding for postsynaptic receptors, ion channels and transporters, cytoskeletal and synaptic proteins (*Bruce et al., 2004*). However, some evidences showed that it may occasionally act as a transcriptional activator (*Kallunki et al., 1998*; *Perera et al., 2015*).

Although downregulated in differentiated neurons, REST levels are dynamically tuned in neurons and can be upregulated in response to kainate-induced seizures in vivo (*Hu et al., 2011*; *McClelland et al., 2014*; *McClelland et al., 2011*; *Palm et al., 1998*; *Spencer et al., 2006*) and chronic hyperactivity in cultured neurons (*Pecoraro-Bisogni et al., 2018*; *Pozzi et al., 2013*). Moreover, REST levels progressively increase in cortical neurons of healthy humans over aging because of the increased Wnt/β-catenin signaling (*Lu et al., 2014*). While the role of REST in epileptogenesis is still debated (*Baldelli and Meldolesi, 2015*; *Garriga-Canut et al., 2006*; *Hu et al., 2011*; *Lignani et al., 2020*; *McClelland et al., 2014*; *McClelland et al., 2011*), its function in aging was demonstrated to be protective from oxidative stress and amyloid β toxicity. Conditional deletion of REST in the mouse brain leads to age-related neurodegeneration and nuclear REST is decreased in mild cognitive impairment and Alzheimer's brains, while elevated REST levels are associated with the preservation of cognitive functions in aged humans (*Lu et al., 2014*). More recently, REST was found to be upregulated in the brain of humans with extended longevity (*Zullo et al., 2019*).

Since neural excitation is believed to increase with age, the capability of REST to repress excitation-related genes (*Pozzi et al., 2013*) suggests that the activation of REST and reduction of excitatory neural activity could be responsible for slowing aging in humans (*Zullo et al., 2019*). These results are consistent with our previous studies, where we demonstrated that the hyperactivity-dependent activation of REST furthers neuronal-network homeostasis by downregulating intrinsic excitability and presynaptic function of excitatory neurons in response to chronic hyperactivity (*Pecoraro-Bisogni et al., 2018*; *Pozzi et al., 2013*). However, whether REST governs homeostatic plasticity by acting also on inhibitory GABAergic interneurons is still unknown.

Here, we demonstrate that REST translocates to the nucleus in excitatory and inhibitory neurons in response to hyperactivity. On inhibitory transmission, this is followed by upscaling of strength and density of perisomatic synapses and downscaling of dendritic synapses only when the postsynaptic target is an excitatory neuron. The increase of the perisomatic inhibitory inputs involves the activation of the P1 promoter-regulated *Bdnf* transcript through a REST-dependent activation of the immediate-early gene (IEG) *Npas4*, suggesting a retrograde action of BDNF released by excitatory neurons on perisomatic inhibitory terminals. The impairment of dendritic inhibition is instead exclusively REST-dependent, suggesting a direct activation of REST in inhibitory neurons.

These findings emphasize the role of REST as a master transcriptional regulator of neuronal homeostasis. By specifically acting at excitatory and inhibitory connections, REST activates a coordinated program that maintains brain circuits activity within physiological levels.

## Results

### Neuronal hyperactivity induces nuclear translocation of REST/NRSF in inhibitory neurons

To study the transcriptional activity of REST in excitatory and inhibitory neurons, primary hippocampal neurons from *Gad1*-GFP mice (*Tamamaki et al., 2003*; *Valente et al., 2016*) were treated with a

Cy3-tagged decoy oligodeoxynucleotide, complementary to the RE1-binding domain of REST (Cy3-ODN), that was chemically modified to improve its stability (*Soldati et al., 2011*) and used to trace endogenous REST. A random decoy oligodeoxynucleotide sequence (Cy3-NEG) served as a negative control. Both Cy3-ODN and Cy3-NEG, added extracellularly, were able to efficiently cross the plasma membrane and diffuse in the cytosol (*Figure 1A and B*; *Figure 1—figure supplement 1A, B*). Interestingly, upon hyperactivity induced by 1 hr treatment with the convulsant 4-aminopyrodine (4AP; 100 µM), Cy3-ODN doubled its partitioning ratio into the nucleus in both GFP-negative excitatory neurons and GFP-positive inhibitory neurons (*Figure 1C–E*). Conversely, Cy3-NEG treated neurons showed a homogeneous diffusion in both nucleus and cytosol that was unaffected by the 4AP treatment (*Figure 1—figure supplement 1*). These results confirm that REST exerts its transcriptional activity through its translocation from the cytoplasm to the nucleus also in GABAergic neurons (*Shimojo, 2008*; *Shimojo et al., 2001*; *Shimojo and Hersh, 2003*).

The nuclear translocation of REST induced by hyperactivity was associated with downregulation of the mRNAs of *Hcn1*, *Syn1*, and *Scn2a* (*Nav1.2*), three well-known REST target genes (*McClelland et al., 2011*; *Paonessa et al., 2013*; *Pozzi et al., 2013*) in control neurons, while it was fully blocked in neurons treated with ODN (*Figure 1—figure supplement 2*). Altogether, these data demonstrate that: (i) ODN is an excellent intracellular tracer of REST; (ii) it inhibits REST activity on its target genes; and (iii) the nuclear translocation of REST upon hyperactivity occurs in both inhibitory and excitatory neurons.

## GABAergic inhibition contributes to the REST-dependent homeostatic recovery from network hyperactivity

To investigate the role played by inhibitory transmission in REST-dependent homeostatic processes, primary cortical networks were treated at 17 days in vitro (div) with either NEG or ODN in the presence or absence of 4AP and their spontaneous firing activity was monitored over time (1–48 hr) by microelectrode array (MEA) recordings (*Figure 2A*). Untreated networks (NEG/veh and ODN/veh) behaved in a similar way, showing a progressive ~20% increase of mean firing rate (MFR), mostly due to an increased burst duration (BD), as previously observed under similar culture conditions (*Pozzi et al., 2013*). Soon after 4AP stimulation (1 hr), NEG-treated neurons displayed greatly enhanced MFR and burst frequency (BF), accompanied by a moderate decrease of BD (*Figure 2B*). Both firing and bursting rates started to decrease after 24 hr of 4AP exposure and were significantly downscaled at the levels of control neurons at 48 hr, while the decrement of BD, acutely induced by 4AP, persisted after 48 hr. Blockade of REST activity by ODN in neurons treated for 48 hr with 4AP did not alter the acute effect of 4AP, but significantly reduced the homeostatic downscaling of both MFR and BF and fully removed the 4AP-induced decrease in BD (*Figure 2B*).

To evaluate the contribution of GABAergic inhibition to the REST-dependent homeostatic response to 4AP, we blocked GABA$_A$ receptors (GABA$_A$ Rs) with bicuculline (BIC, 30 µM) at the end of the 48 hr treatment with 4AP. While BD was similarly increased by BIC in all experimental groups, the GABA$_A$R blockade induced a significantly larger increase in MFR and BF in NEG-treated neurons stimulated with 4AP than in NEG-treated neurons stimulated with vehicle (*Figure 2C*), suggesting the removal of a synaptic inhibition whose strength was increased by hyperactivity. Interestingly, such effect was abolished in ODN-treated neurons, demonstrating an involvement of REST in the upscaling of GABAergic transmission.

## Hyperactivity induces a REST-mediated increase of mIPSC frequency in excitatory, but not inhibitory, neurons

Miniature postsynaptic currents (mPSCs) are widely used to get information about changes in synaptic properties during homeostatic plasticity (*Turrigiano et al., 1998*; *Turrigiano, 2008*; *Pecoraro-Bisogni et al., 2018*). While changes in amplitude mostly depend on the density or conductance of postsynaptic receptors (*O'Brien et al., 1998*), changes in frequency reflect the number of active synapses and/or the quantal release probability (*Shao and Dudek, 2005*). To evaluate whether GABAergic transmission is effectively enhanced in 4AP-treated neurons, we measured miniature inhibitory postsynaptic currents (mIPSCs) in excitatory (GFP-negative) and inhibitory (GFP-positive) neurons treated with NEG/vehicle, NEG/4AP, ODN/vehicle, and ODN/4AP for 48 hr (*Figure 3A and E*). Chronic treatment with 4AP significantly increased the frequency of mIPSCs recorded from excitatory neurons, an

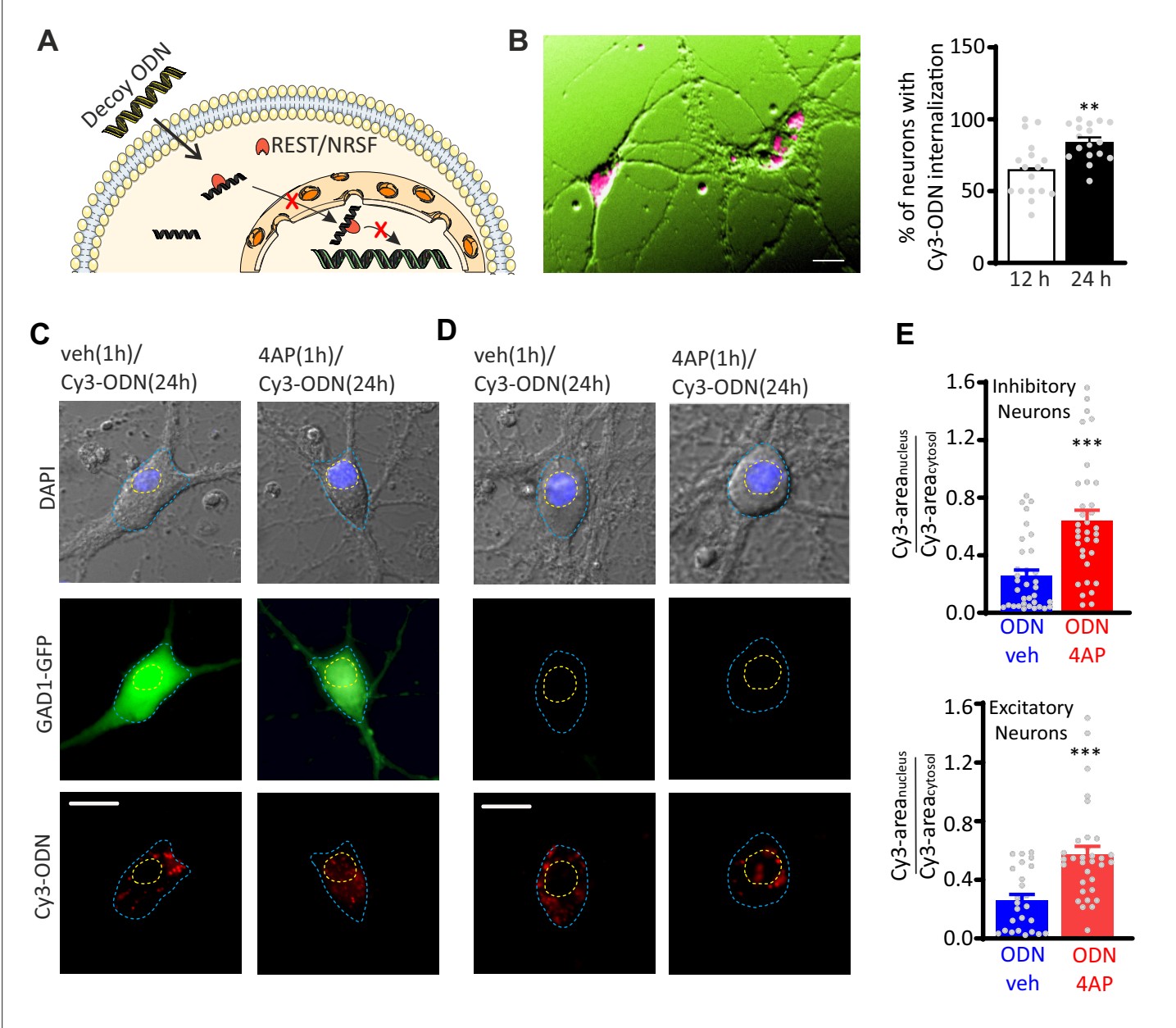

**Figure 1.** Hyperactivity induces REST translocation to the nucleus in both inhibitory and excitatory neurons. (**A**) Schematic representation of the ODN action. Having a complementary sequence to RE1, ODN binds REST, thus abrogating its capability to modulate its transcriptional activity. (**B**) *Left:* Primary hippocampal neurons (16–18 div) were incubated for 12 and 24 hr with Cy3-tagged ODN (Cy3-ODN, 200 nM). *Right:* Cy3-ODN efficiently permeated into primary hippocampal neurons. The bar plot shows the mean (± SEM) percentage of neurons that internalized Cy3-ODN after 12 and 24 hr of incubation (n=17 fields from two independent preparations; **p<0.01; unpaired Student's t-test). (**C–E**) Tracking of Cy3-ODN reveals both cytoplasmic and nuclear localizations of REST and its nuclear translocation upon hyperactivity in both excitatory and inhibitory neurons. (**C, D**) Representative fluorescence images of hippocampal GFP-positive inhibitory (*left*) and GFP-negative excitatory (*right*) neurons (17 div) treated with either vehicle or 4AP for 1 hr and subsequently labeled with Cy3-ODN for 24 hr. (**E**) Bar plots represent the mean± SEM of the REST partition ratio between cytosol and nucleus (Cy3-area$_{nucleus}$/Cy3-area$_{cytosol}$) in inhibitory (*upper panel*; ODN/veh n=33; ODN/4AP n=34) and excitatory (*lower panel*; ODN/veh n=25; ODN/4AP=32) neurons. ***p<0.001; Mann–Whitney U-test. Scale bars: 20 µm (**B**), 10 µm (**C, D**). Data for (**B**) and (**E**) can be found in *Figure 1—source data 1*.

The online version of this article includes the following figure supplement(s) for figure 1:

**Source data 1.** Source data for *Figure 1*.

**Figure supplement 1.** A scrambled version of ODN (Cy3-NEG) does not translocate to the nucleus on hyperactivity.

**Figure supplement 2.** Treatment with ODN blocks the repression of REST target genes upon hyperactivity.

*Figure 1 continued on next page*

*Figure 1 continued*

**Figure supplement 2—source data 1.** Source data for *Figure 1—figure supplement 2*.

effect that was fully blocked by ODN (*Figure 3B and C*). Conversely, 4AP or ODN treatment did not affect the mIPSC amplitude (*Figure 3B and D*). Strikingly, 4AP-mediated chronic hyperactivity had no effects on both mIPSC frequency and amplitude recorded from inhibitory neurons (*Figure 3F–H*).

These results indicate the presynaptic origin of the REST-dependent homeostatic changes in inhibitory transmission in response to hyperactivity and demonstrate that these effects are fully target-specific, only occurring in inhibitory-to-excitatory neuron synapses.

## Neuronal hyperactivity induces a REST-mediated increase of evoked IPSCs onto excitatory neurons

To investigate whether the increased mIPSC frequency recorded in excitatory neurons is attributable to a change in the number of active inhibitory synapses or in the quantal release probability at existing sites, we investigated evoked postsynaptic inhibitory currents (eIPSCs) in 18–21 div hippocampal neurons. Extracellular minimal stimulation in loose-patch configuration was used to evoke action potentials (APs) in GFP-positive presynaptic inhibitory interneurons and patch-clamp recordings were obtained from postsynaptic GFP-negative excitatory neurons (*Figure 4A*). Evoked IPSCs in excitatory neurons treated for 48 h with 4AP displayed an increased amplitude that was suppressed upon treatment with ODN. The 4AP-induced eIPSC enhancement was accompanied by a decrease in the paired-pulse ratio (PPR), which was also abolished by treatment with ODN (*Figure 4B*).

To better define the mechanism by which neuronal hyperactivity modulates GABA release, we estimated the readily releasable pool for synchronous release (RRP$_{syn}$) and the probability of release (P$_r$) of any given synaptic vesicle in the RRP, using the cumulative amplitude analysis (*Baldelli et al., 2005*; *Schneggenburger et al., 1999*). When inhibitory synapses onto excitatory neurons were challenged with a 2 s train at 20 Hz (40 APs), a significant depression of eIPSCs became apparent during the stimulation period, irrespective of the amplitude of the first current of the train. In NEG-treated neurons, 4AP increased synaptic depression, an effect that was virtually absent in ODN-treated cells (*Figure 4C and D*). We next analyzed the cumulative profiles of eIPSC amplitude that display a rapid rise followed by a slower linear increase reflecting the equilibrium between depletion and constant replenishment of the RRP$_{syn}$ (*Figure 4E*). The graphical extraction of the RRP$_{syn}$ and P$_r$ from the cumulative curves of individual neurons showed that the 4AP-induced eIPSC increase was due to an increase in both RRP$_{syn}$ and P$_r$, while both quantal parameters were not affected by 4AP in ODN-treated neurons (*Figure 4F and G*).

Interestingly, the eIPSC amplitude and PPR recorded from inhibitory interneurons revealed that the synaptic strength of inhibitory synapses onto excitatory neurons was not modulated by 4AP. Accordingly, the cumulative amplitude analysis confirmed that the quantal parameters RRP$_{syn}$ and P$_r$ were unaffected by chronic neuronal hyperactivity and/or ODN treatment (*Figure 4—figure supplement 1*). The quantal analysis of synaptic strength and short-term plasticity of inhibitory transmission in neurons exposed to chronic hyperactivity confirmed the striking postsynaptic target-specificity of the homeostatic adaptations, as already highlighted by the mIPSC analysis, demonstrating that the REST-dependent upscaling of inhibitory transmission only occurs when the postsynaptic target is an excitatory neuron.

## REST-dependent increase of perisomatic GABAergic synapses onto excitatory neurons

Although the increased mIPSC frequency and eIPSC amplitude directly correlated with parallel changes in P$_r$ and RRP$_{syn}$ size, a concomitant increase in the number of functional synaptic contacts cannot be excluded. Moreover, being patch-clamp mostly sensitive to synaptic currents of somatic origin, the functional changes observed in GABAergic synapses onto excitatory neurons predominantly regard perisomatic synapses. To evaluate whether the REST-mediated homeostatic response to hyperactivity also affects the density of GABAergic synapses at somatic level and in more peripheral dendritic districts, we triple immunostained hippocampal neurons for VGAT, gephyrin, and β3-tubulin to label inhibitory presynaptic terminals, inhibitory postsynaptic scaffolds, and somato-dendritic structures,

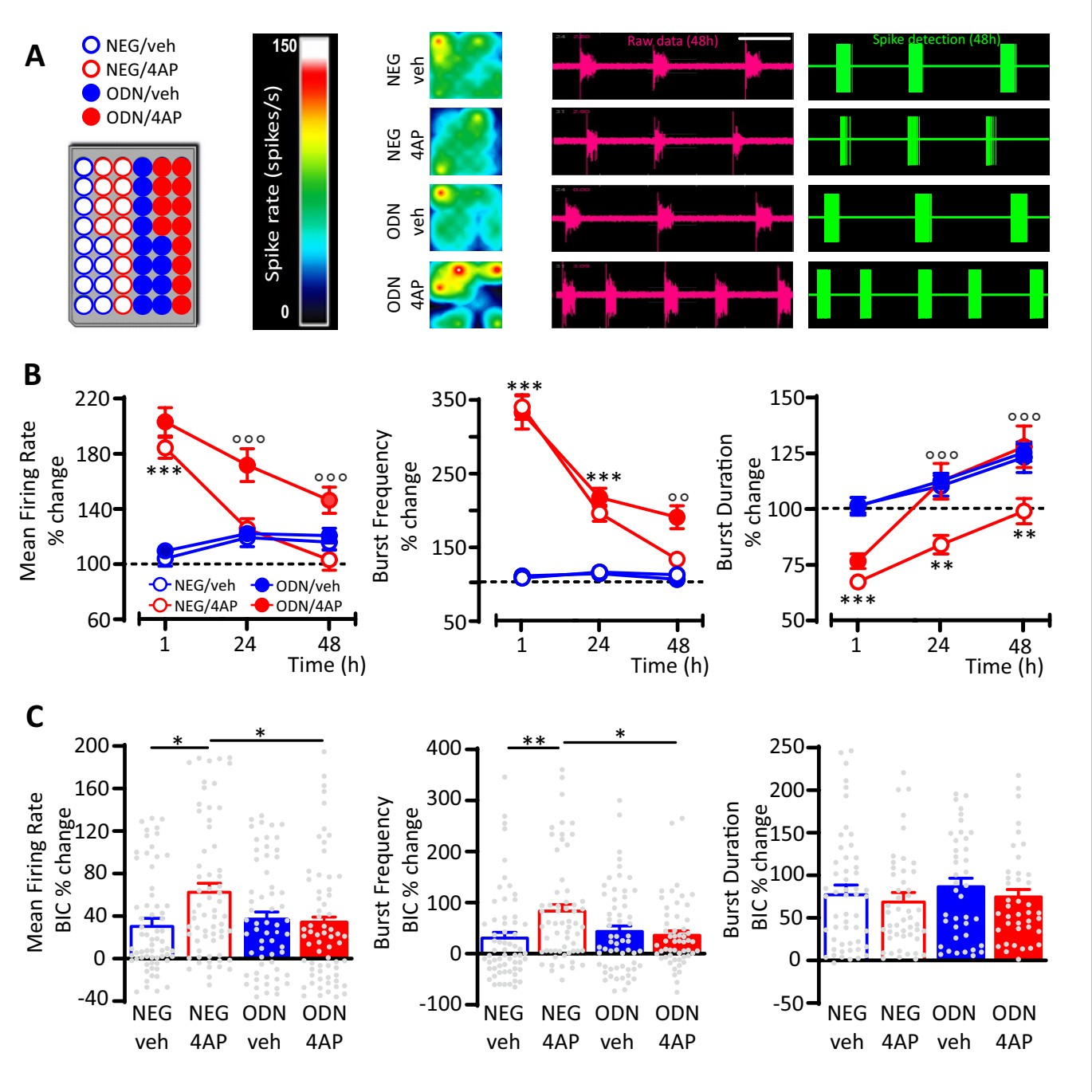

**Figure 2.** GABAergic transmission contributes to the REST-dependent homeostatic response to hyperactivity. (**A**) *Left panel:* Representative MEA plate-map showing treatment group assignment (NEG/vehicle; NEG/4AP; ODN/vehicle; ODN/4AP). *Right panels:* Firing rate heat maps (*left*), raw voltage recordings from individual electrodes (*middle*) and corresponding spike detection results (*right*), for representative wells of the four experimental groups maintained for 48 hr in 4AP. (**B**) Mean firing rate (MFR) (*left*), bursting rate (*middle*) and burst duration (BD) (*right*) are expressed in percent of the NEG/veh baseline as a function of time after the onset of the treatments (1, 24, and 48 hr). **p<0.01, ***p<0.001 NEG/veh versus NEG/4AP; °°p<0.01, °°°p<0.001 NEG/4AP versus ODN/4AP; two-way ANOVA/Tukey's tests. (**C**) Percent increases in MFR (*left*), burst frequency (*middle*), and BD (*right*) induced by BIC (30 µM) after 48 hr of the indicated treatments. *p<0.05, **p<0.01; two-way ANOVA/Tukey's tests. Data are means± SEM (panel (**B**), 78<n<59; panel (**C**), 65<n<45 MEA wells for NEG/veh, NEG/4AP; ODN/veh and ODN/4AP, from n=7 independent neuronal preparations). Data for (**B**) and (**C**) can be found in *Figure 2—source data 1*.

The online version of this article includes the following figure supplement(s) for figure 2:

**Source data 1.** Source data for *Figure 2*.

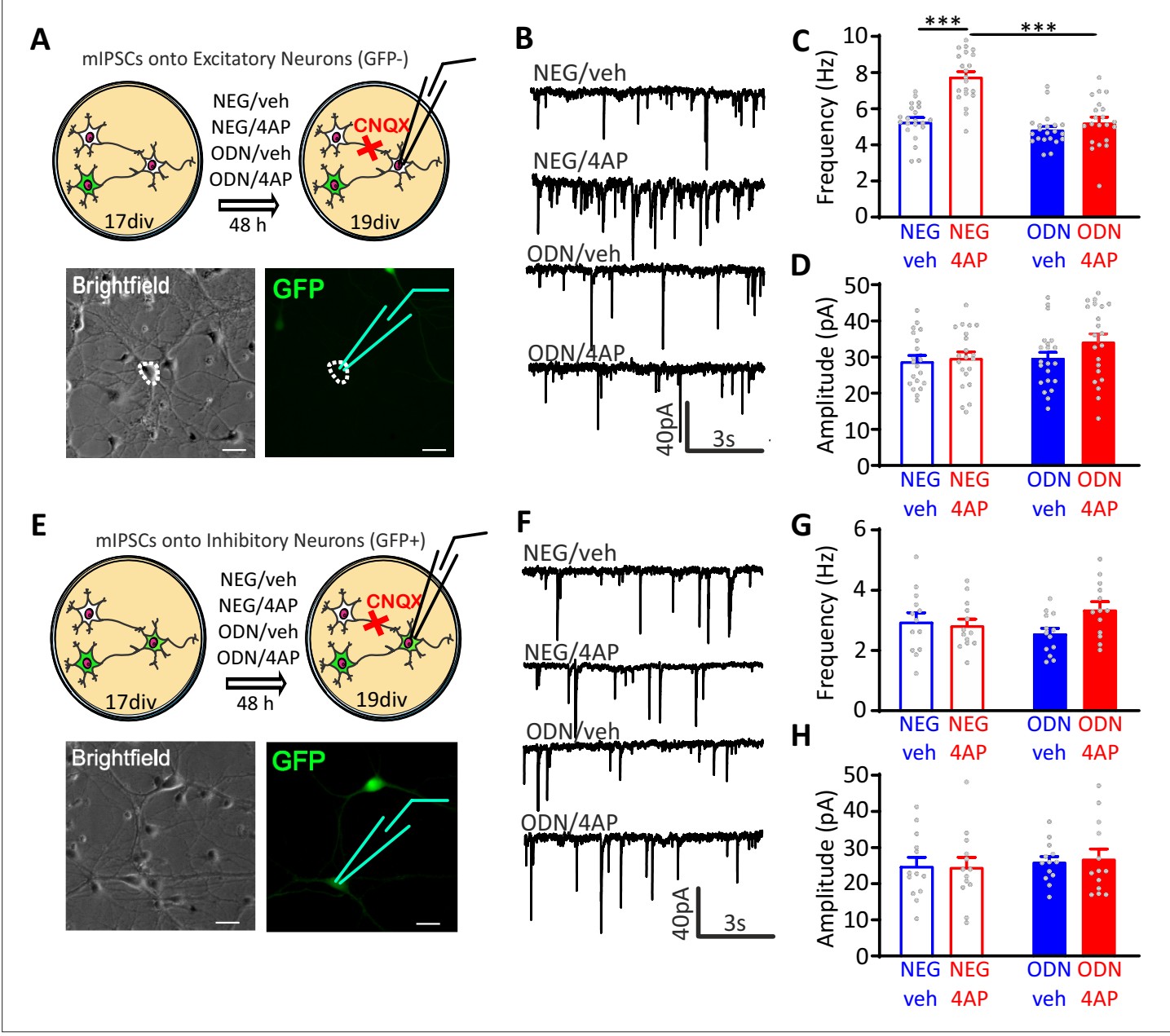

**Figure 3.** Neuronal hyperactivity selectively increases the frequency of mIPSCs in excitatory neurons in a REST-dependent fashion. (**A**) Schematic representation and representative microphotographs of a patched GFP-negative excitatory neuron. (**B–D**) Representative mIPSC traces (**B**) recorded at 19 div in the four experimental groups after 48 hr of treatment. Means± SEM of mIPSC frequency (**C**) and amplitude (**D**) of NEG/vehicle (n=21), NEG/4AP (n=21), ODN/vehicle (n=21), and ODN/4AP (n=21) treated neurons. (**E**) Schematic representation and representative microphotograph of a patched GFP-positive inhibitory neuron. (**F–H**) Representative mIPSC traces (**F**) recorded at 19 div in the four experimental groups after 48 hr of treatment. Mean (± SEM) IPSC frequency (**G**) and amplitude (**H**) of NEG/vehicle (n=13), NEG/4AP (n=13), ODN/vehicle (n=13), and ODN/4AP (n=13) treated neurons. ***p<0.001, two-way ANOVA/Tukey's tests. Scale bars, 30 μm. Data for (**C**), (**D**), (**G**), and (**H**) can be found in *Figure 3—source data 1*. mIPSC, miniature inhibitory postsynaptic current.

The online version of this article includes the following figure supplement(s) for figure 3:

**Source data 1.** Source data for *Figure 3*.

respectively (*Figure 5A*, *Figure 5—figure supplement 1A*). We then analyzed inhibitory synapses along the soma and dendrites of GFP-negative excitatory neurons and GFP-positive inhibitory neurons. When the postsynaptic target neuron was excitatory, the sustained hyperactivity doubled the density of inhibitory perisomatic synapses and slightly, but significantly, decreased dendritic contacts. The

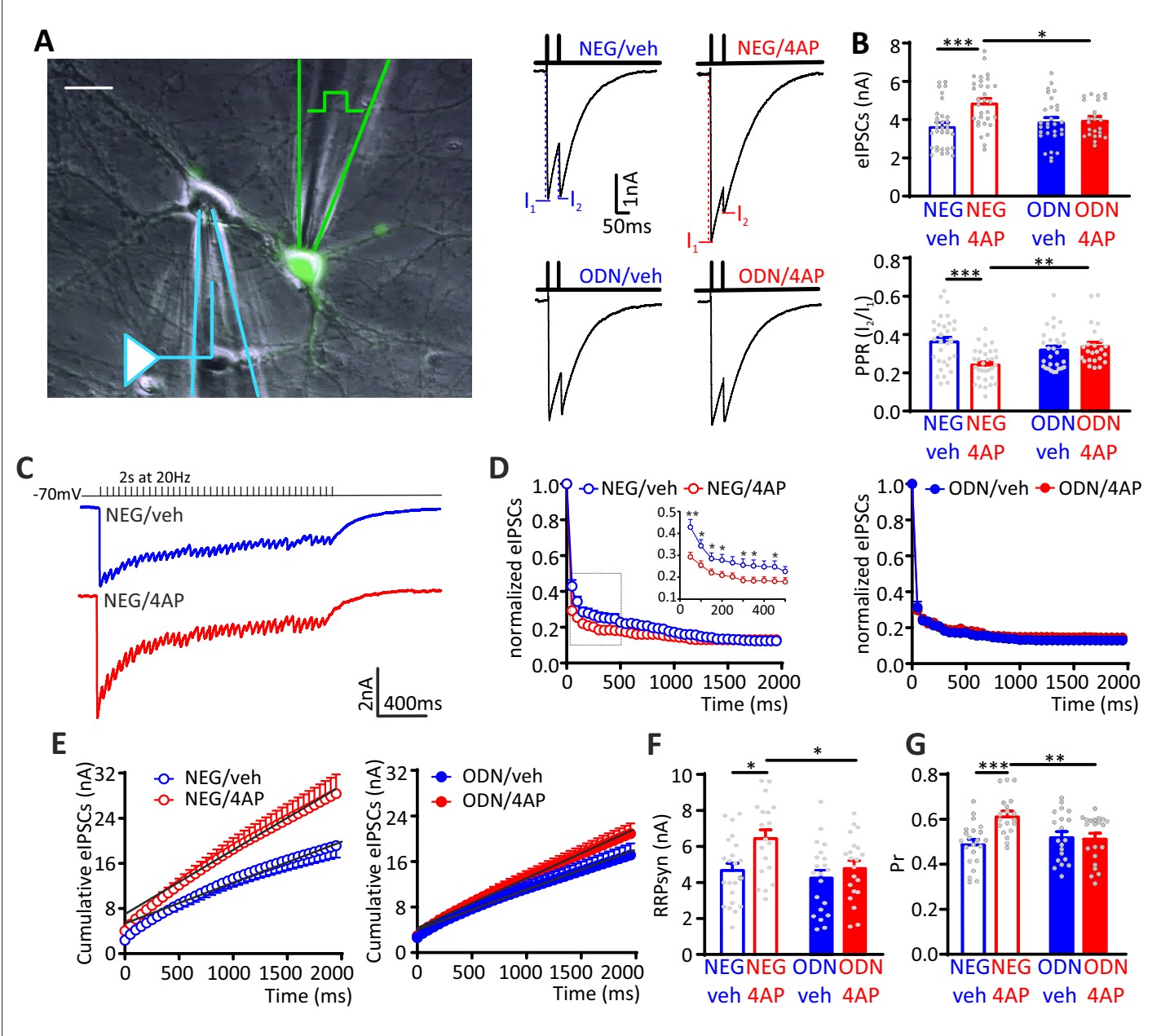

**Figure 4.** The hyperactivity-induced upscaling of eIPSCs is specific for excitatory neurons and mediated by a REST-dependent increase in RRP$_{syn}$ and P$_r$. (**A**) *Left:* Experimental setup showing the stimulation electrode located on a GFP-positive GABAergic neuron and the recording electrode patching a GFP-negative excitatory neuron. Scale bar, 20 μm. *Right:* Representative eIPSCs evoked by a paired-pulse stimulation protocol (interpulse interval, 50 ms). (**B**) The amplitude of the first eIPSC in the pair (*top*) and the paired-pulse ratio (PPR=I$_2$/I$_1$; *bottom*) of NEG/vehicle (n=32), NEG/4AP (n=31), ODN/vehicle (n=29), or ODN/4AP (n=24) treated neurons are shown as means± SEM. (**C–G**) Quantal analysis of RRP$_{syn}$ and P$_r$ in GABAergic synapses onto excitatory neurons. (**C**) Representative eIPSC traces evoked by a 2-s tetanic stimulation at 20 Hz in NEG-treated neurons in the absence (blue) or presence (red) of 4AP. (**D**) Averaged plot of normalized eIPSC amplitude versus time during the tetanic stimulation. The inset shows the boxed area in an expanded time scale. (**E**) Averaged cumulative profiles of eIPSCs. To calculate RRP$_{syn}$, data points in the 1–2 s range were fitted by linear regression and back extrapolated to time 0. (**F, G**) Means (± SEM) of the individual values of RRP$_{syn}$ (**F**) and P$_r$ (**G**) of NEG/vehicle (n=22), NEG/4AP (n=19), ODN/vehicle (n=21), or ODN/4AP (n=21) treated neurons. *p<0.05, **p<0.01, ***p<0.001; two-way ANOVA/Tukey's tests. Data for (**B**), (**D**), (**E**), (**F**), and (**G**) can be found in *Figure 4—source data 1*. eIPSC, evoked postsynaptic inhibitory current.

The online version of this article includes the following figure supplement(s) for figure 4:

**Source data 1.** Source data for *Figure 4*.

**Figure supplement 1.** eIPSCs recorded in inhibitory interneurons are not homeostatically modulated by neuronal hyperactivity.

**Figure supplement 1—source data 1.** Source data for *Figure 4—figure supplement 1*.

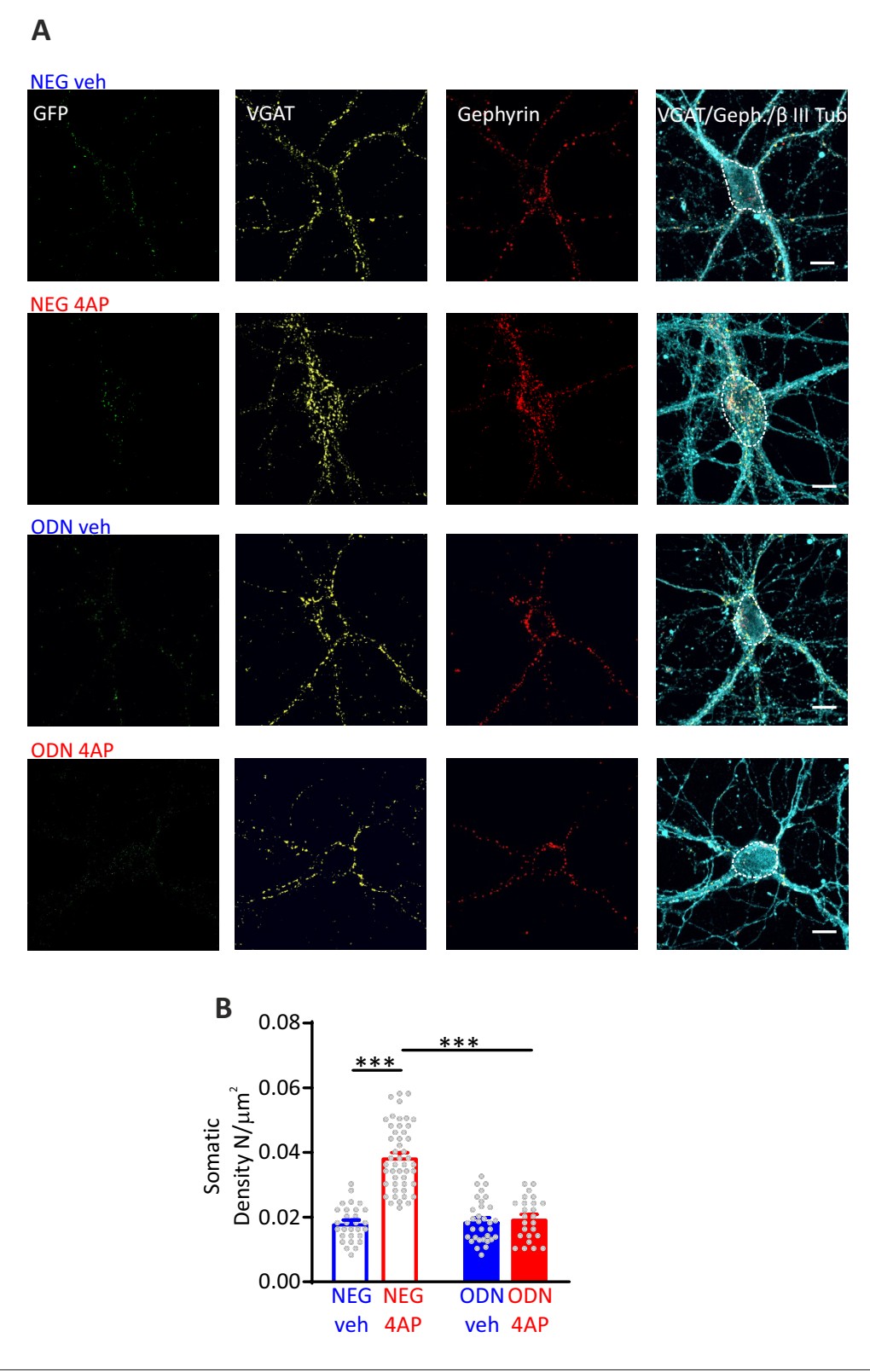

**Figure 5.** Hyperactivity induces a REST-dependent specific increase of perisomatic inhibitory synapses onto excitatory neurons. (**A**) Representative microphotographs showing GFP-negative excitatory neurons (20 div), treated with NEG/ctrl, NEG/4AP, ODN/veh, and ODN/4AP, labeled with β3-tubulin (light blue) and decorated with gephyrin (red) and VGAT (yellow) antibodies to identify GABAergic synapses. White lines highlight somatic areas.

*Figure 5 continued on next page*

*Figure 5 continued*

Scale bars, 10 µm. (**B**) Quantification of the density of somatic inhibitory synapses onto excitatory neurons. Data are means± SEM of 48<n<25, respectively, from three independent neuronal preparations. ***p<0.001; two-way ANOVA/Tukey's tests. Data for (**B**) can be found in *Figure 5—source data 1*.

The online version of this article includes the following figure supplement(s) for figure 5:

**Source data 1.** Source data for *Figure 5*.

**Figure supplement 1.** Hyperactivity induces a REST-dependent decrease of dendritic inhibitory synapses onto excitatory neurons.

**Figure supplement 1—source data 1.** Source data for *Figure 5—figure supplement 1*.

**Figure supplement 2.** Hyperactivity does not affect the density of inhibitory synapses onto inhibitory neurons.

**Figure supplement 2—source data 1.** Source data for *Figure 5—figure supplement 2*.

changes of both perisomatic and dendritic inhibitory synapses required an intact REST transcriptional activity, as they were fully blocked by treatment with ODN (*Figure 5B*, *Figure 5—figure supplement 1B*). The strict target specificity of these effects was further underlined by the absence of any change on both somatic and dendritic inhibitory synapses when the postsynaptic target was an inhibitory neuron (*Figure 5—figure supplement 2*).

## Inhibition of BDNF-TrkB binding mimics the effect of REST blockade on the upscaling of inhibitory inputs onto excitatory neurons

It is widely accepted that BDNF is expressed and released in an activity-dependent fashion primarily by excitatory neurons (*Canals et al., 2001*; *Dieni et al., 2012*; *Ernfors et al., 1990*; *Matsumoto et al., 2008*; *Hofer et al., 1990*; *Isackson et al., 1991*) while it is absent from cortical inhibitory inter-neurons (*Cohen-Cory et al., 2010*; *Andreska et al., 2014*; *Spiegel et al., 2014*). In addition, BDNF has a well-known action on the development of GABAergic synapses (*Huang et al., 1999*; *Marty et al., 2000*; *Seil and Drake-Baumann, 2000*; *Yamada et al., 2002*; *Ohba et al., 2005*; *Lin et al., 2008*) that mostly occurs presynaptically (*Baldelli et al., 2005*; *Baldelli et al., 2002*) at perisomatic synapses (*Fiorentino et al., 2009*; *Jiao et al., 2011*).

Based on this evidence, we focused on a possible crosstalk between REST and BDNF to elucidate molecular mechanisms of the target-specific REST action on inhibitory transmission. We first analyzed the time-course of the changes in *Rest* and *coding sequence-Bdnf* (*cds-Bdnf*), mRNA levels in neurons treated with 4AP for 6, 24, 48, and 96 hr (*Figure 6A*). We also included the mRNA of the membrane trafficking protein Synaptotagmin-4 (SYT4), known to localize in BDNF-containing vesicles and modulate their release (*Dean et al., 2009*). A significant increase in *Rest*, *cds-Bdnf*, and *Syt4* mRNA levels was already apparent after 6 hr of treatment with 4AP. While the *Rest* elevation persisted for 48 hr and recovered to basal level only after 96 hr, the fast increments in *cds-Bdnf* and *Syt4* mRNAs were short-lasting and fully recovered their basal values after 48 hr (*Figure 6A*).

To investigate the molecular relationship between REST and BDNF transcription, we compared the hyperactivity-dependent increase of *Rest* and *Bdnf* mRNAs in control and *Rest* knockdown (KD) hippocampal neurons (*Figure 6B*). While the hyperactivity-dependent increase of *Rest* mRNA was fully suppressed by R*Rest* KD (*Pecoraro-Bisogni et al., 2018*; *Pozzi et al., 2013*), the temporal profile of *cds-Bdnf* mRNA induced by hyperactivity was markedly altered by *Rest* silencing. REST deficiency significantly reduced the transient increase of *Bdnf* at 24 hr, while it completely suppressed the recovery of *Bdnf* mRNA to basal levels at 48 hr, transforming the bell-shaped *Bdnf* expression curve into a monotonic increment.

To evaluate whether BDNF could contribute to the REST-dependent upscaling of inhibitory synapses, the effect of 4AP on evoked inhibitory transmission was studied in hippocampal excitatory neurons (18 div) treated for 48 hr with either vehicle (ctrl) or the BDNF-scavenger TrkB-fc (TrkB-fc), a recombinant chimeric protein that suppresses BDNF binding to TrkB receptors (*Sakuragi et al., 2013*; *Figure 6C*). Interestingly, the previously observed increase of eIPSC amplitude and decrease of PPR evoked by 4AP were fully suppressed by BDNF sequestration (*Figure 6D*).

In line with this result, the 4AP-induced enhancement of perisomatic GABAergic contacts onto excitatory neurons was also fully suppressed in neurons treated with TrkB-fc (*Figure 7*). On the contrary, the 4AP-induced decrease of dendritic GABAergic contacts to excitatory neurons was insensitive to

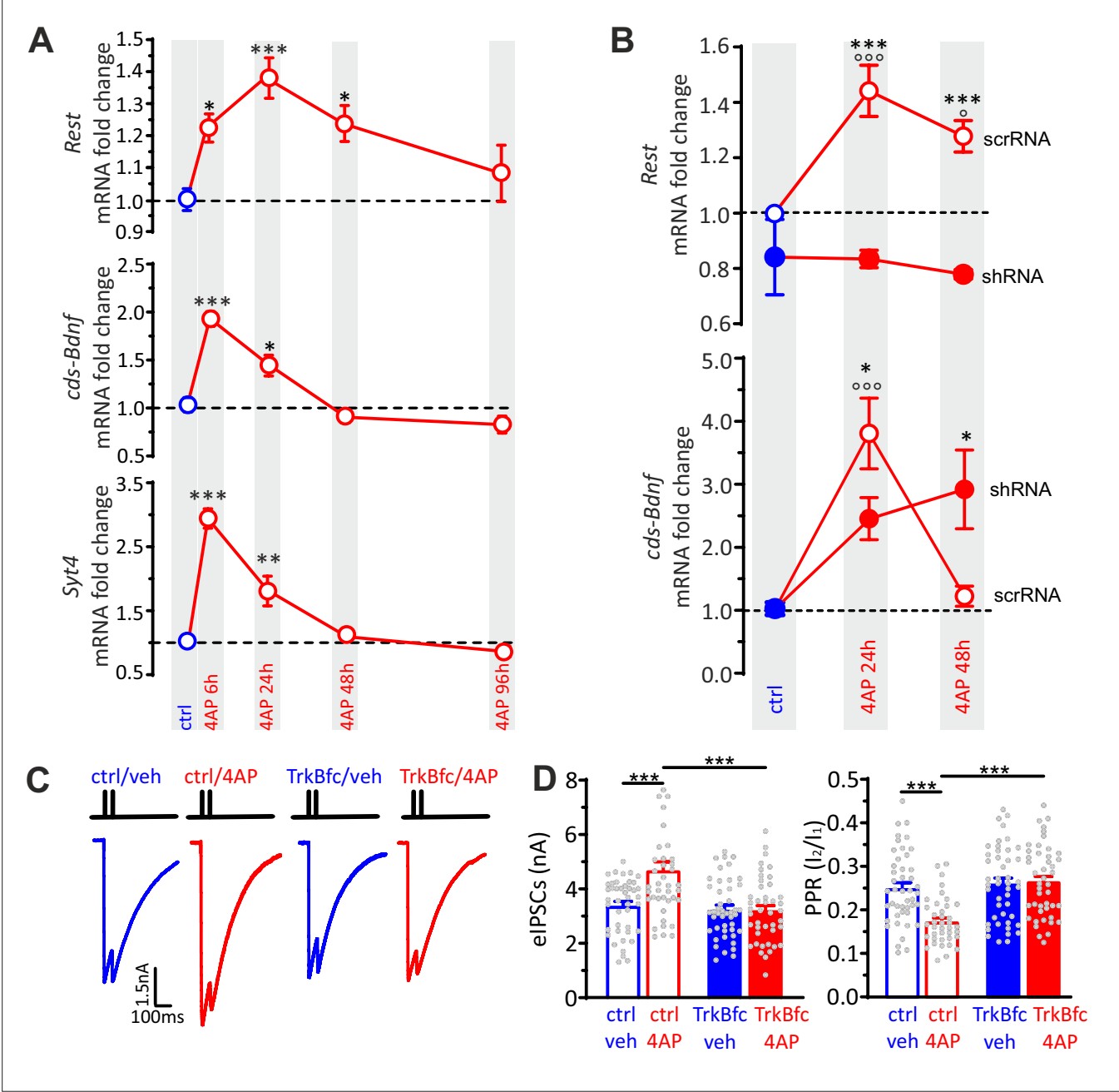

**Figure 6.** BDNF specifically potentiates GABAergic synapses onto excitatory neurons in response to hyperactivity. (**A**) Time-course of *Rest*, *cds-Bdnf*, and *Syt4* mRNA fold changes in control (blue symbols) and 4AP-treated (red symbols) cortical neurons treated for various times with 4AP. All 4AP-treated samples were collected at 20 div after 6, 24,48, and 96 hr of 4AP treatment, respectively. The control sample was also collected at 20 div without any prior treatment. For each time point, data are means± SEM of 9<n<10 from three independent neuronal preparations. *p<0.05, **p<0.01, ***p<0.001 versus control; one-way ANOVA/Dunnett's tests. (**B**) RT-qPCR analysis of changes (means± SEM) of *Rest* and *cds-Bdnf* mRNA transcript levels in control (blue symbols) and 4AP-treated (red symbols; 24 and 48 hr) neurons that had been infected with either scrRNA (*open symbols*) or shRNA (*closed symbols*) viruses. Treatments were performed as described in (**A**), with neurons collected at 20 div. For each time point, data are means± SEM of 9<n<10 from three independent neuronal preparations. *p<0.05, ***p<0.001 shRNA versus scrRNA; °p<0.05, °°°p<0.001 4AP-treated versus control; two-way ANOVA/Tukey's tests. (**C**) Representative eIPSCs onto GFP-negative excitatory neurons in response to paired-pulse stimulation recorded in ctrl/veh, ctrl/4AP, TrkB-fc/veh and TrkB-fc/4AP treated neurons. (**D**) Mean (± SEM) amplitude of the first eIPSC ($I_1$; left) and paired-pulse ratio (PPR=$I_2/I_1$; right) of ctrl/veh (n=43), ctrl/4AP (n=39), TrkB-fc/veh (n=44), and TrkB-fc/4AP (n=43) treated neurons. *p<0.05, ***p<0.001, two-way ANOVA/Tukey's tests. Data for (**A**) and (**B**) can be found in *Figure 6—source data 1*.

The online version of this article includes the following figure supplement(s) for figure 6:

**Source data 1.** Source data for *Figure 6*.

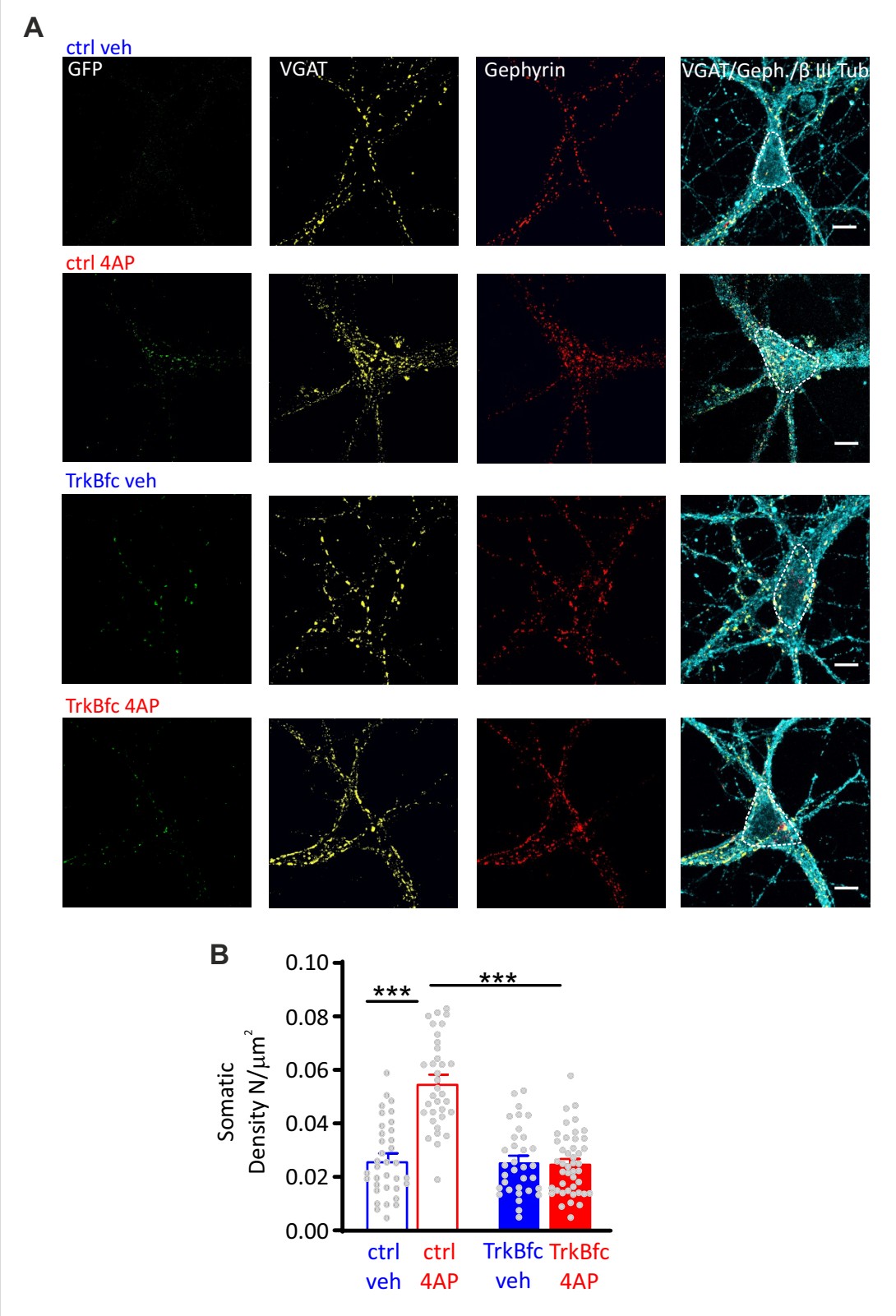

**Figure 7.** BDNF scavenging suppresses the 4AP-induced enhancement of perisomatic inhibitory synapses to excitatory neurons. (**A**) Representative microphotographs showing GFP-negative excitatory neurons (20 div), treated with ctrl/veh, ctrl/4AP, TrkB-fc/veh, and TrkB-fc/4AP, labeled with β3-tubulin (light blue) and decorated with gephyrin (*red*) and VGAT (*yellow*) antibodies to identify GABAergic synapses. White lines highlight somatic areas. Scale bars, 10 μm. (**B**) Quantification (means± SEM) of the densities of somatic inhibitory synapses onto excitatory neurons (33<n<44), from two independent

*Figure 7 continued on next page*

*Figure 7 continued*

neuronal preparations. ***p<0.001; two-way ANOVA/Tukey's tests. Data for (**B**) can be found in *Figure 7—source data 1*.

The online version of this article includes the following figure supplement(s) for figure 7:

**Source data 1.** Source data for *Figure 7*.

**Figure supplement 1.** BDNF scavenging does not affect the 4AP-induced reduction of dendritic inhibitory synapses to excitatory neurons.

**Figure supplement 1—source data 1.** Source data for *Figure 7—figure supplement 1*.

**Figure supplement 2.** The lack of effect of 4AP on the density of inhibitory synapses onto inhibitory neurons is not affected by BDNF.

**Figure supplement 2—source data 1.** Source data for *Figure 7—figure supplement 2*.

**Figure supplement 3.** BDNF is a downstream effector of REST for the increase in perisomatic inhibitory synapses onto excitatory neurons induced by chronic hyperactivity.

**Figure supplement 3—source data 1.** Source data for *Figure 7—figure supplement 3*.

BDNF sequestration (*Figure 7—figure supplement 1*). Finally, as previously shown in neurons treated with ODN, either 4AP-induced hyperactivity or BDNF scavenging did not affect the density of both perisomatic and dendritic inhibitory boutons when the postsynaptic target was an inhibitory neuron (*Figure 7—figure supplement 2*).

## The increase of perisomatic synapses depends on the sequential activation of REST and BDNF

Altogether, the results reveal a REST/BDNF-dependent enhancement of strength and density of perisomatic inhibitory synapses onto excitatory neurons, accompanied by a REST-dependent, but BDNF-independent, downscaling of dendritic synapses. To obtain evidence showing that the effects of REST on perisomatic inhibitory transmission depend on BDNF, we further studied the change in perisomatic synapses by combining REST inhibition with either TrkB inhibition or TrkB stimulation. This experiment was aimed at understanding whether the effects of REST on perisomatic inhibition were mediated by a sequential activation of BDNF or if, instead, REST and BDNF worked in parallel through two distinct pathways converging on the regulation of perisomatic synapses. The results (*Figure 7—figure supplement 3*) confirmed that the 4AP-induced increase of perisomatic synapses was fully blocked by either REST inhibition or TrkB inhibition or both, indicating a full occlusion of the homeostatic effect. Importantly, the increase of perisomatic contacts was fully recovered by TrkB activation with exogenous BDNF, irrespective of whether REST was inhibited or not. Interestingly, under conditions of 4AP-induced hyperactivity, treatment with BDNF alone did not further increase the magnitude of the homeostatic change in perisomatic synapses, indicating the lack of additive effects between REST and BDNF.

Altogether, these results demonstrate that the signaling pathway underlying the hyperactivity-induced increase in perisomatic synapses depends on a serial activation of REST and BDNF, excluding the possibility of two independent pathways stimulated by hyperactivity and converging on the same target.

## REST triggers the activation of the "*Npas4-Bdnf*" gene program

We have shown that REST exerts a temporal constraint on the increase of *Bdnf* mRNA, allowing its full recovery to basal levels after 48 hr (see *Figure 6B*). This effect was expected and widely supported by experimental evidence showing that *Bdnf* is a REST-target gene (*Garriga-Canut et al., 2006*; *Paonessa et al., 2016*; *Hara et al., 2009*; *Otto et al., 2007*; *Bruce et al., 2004*). On the contrary, the increase of *Bdnf* mRNA observed in neurons treated for 24 hr with 4AP was significantly reduced in REST KD neurons, demonstrating that REST plays an unexpected role in the early increase of *Bdnf* mRNA induced by hyperactivity (see *Figure 6B*). Searching for a possible causal link between REST and BDNF, we focused our attention on NPAS4, a transcriptional activator recently demonstrated to induce a hyperactivity-induced increase in the number and strength of perisomatic inhibitory synapses onto excitatory neurons by inducing BDNF release from excitatory neurons (*Bloodgood et al., 2013*; *Lin et al., 2008*; *Spiegel et al., 2014*).

The existence of a crosstalk between REST, NPAS4, BDNF, and SYT4 was investigated by analyzing the changes in their transcripts in cultured neurons treated with NEG/vehicle, NEG/4AP, ODN/vehicle,

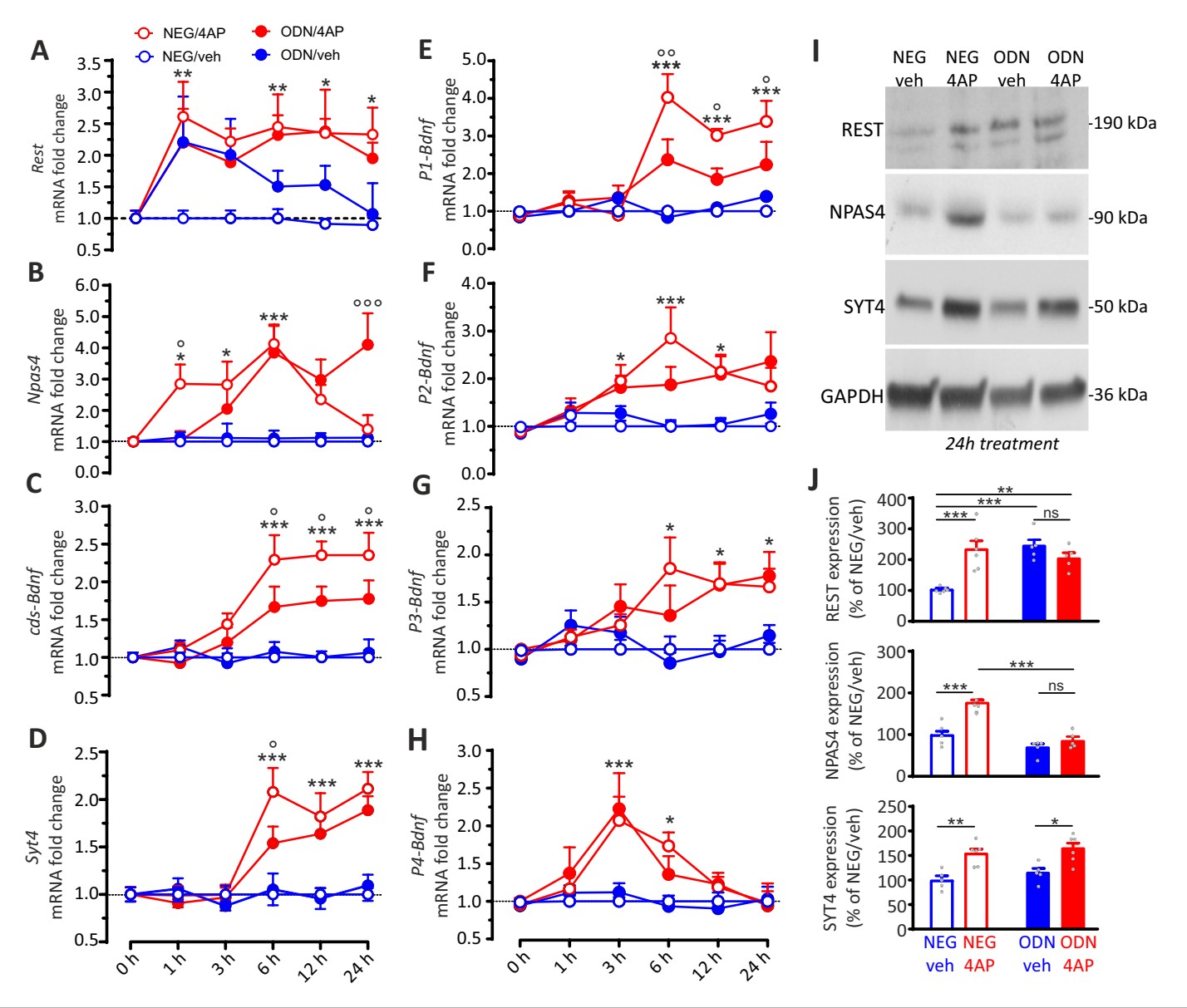

**Figure 8.** The REST-dependent potentiation of GABAergic synapses involves *Npas4* and *P1-Bdnf* activation. (**A–H**) Time course of the REST-dependent transcriptomic profile induced by hyperactivity. Mean (± SEM) fold changes of *Rest* (**A**), *Npas4* (**B**), *cds-Bdnf* (**C**), *Syt4* (**D**), *P1-Bdnf* (**E**), *P2-Bdnf* (**F**), *P3-Bdnf* (**G**), and *P4-Bdnf* (**H**) mRNAs in cortical neurons treated with NEG/vehicle, NEG/4AP, ODN/vehicle, or ODN/4AP before and at various times after the respective treatments. All values are normalized to the NEG/vehicle mRNA levels. For each time point, 11<n<5 from three independent neuronal preparations. *p<0.05, **p<0.01, ***p<0.001, NEG/veh versus NEG/4AP; °p<0.05, °°p<0.01, °°°p<0.001, NEG/4AP versus ODN/4AP; two-way ANOVA/Tukey's tests. (**I, J**) Representative immunoblots (**I**) and quantitative analysis (**J**) of REST, NPAS4 and SYT4 protein expression in cortical neurons treated for 24 hr with NEG/vehicle, NEG/4AP, ODN/vehicle or ODN/4AP. All values (means ± sem) are normalized to the NEG/vehicle level. GAPDH was included as control of equal loading. For each protein, 6<n<5 from three independent neuronal preparations. *p<0.05, **p<0.01; ***p<0.001, two-way ANOVA/Tukey's test. Data for (**A–J**) can be found in *Figure 8—source data 1*.

The online version of this article includes the following figure supplement(s) for figure 8:

**Source data 1.** Source data for *Figure 8*.

and ODN/4AP at a higher temporal resolution (0, 1, 3, 6, 12, and 24 hr). Interestingly, the 4AP-induced increase of *Rest* mRNA was very fast, peaking at 1 hr and persisting for 24 hr (*Figure 8A*). ODN treatment per se, rapidly increased *Rest* mRNA levels under basal conditions (*Johnson et al., 2007*), indicating that the basal levels of REST protein exert a tonic transcriptional repression on the *Rest* gene.

As previously reported (*Lin et al., 2008*), *Npas4* mRNA raised already 1 hr after 4AP, but the increase was transient, and the levels started to decrease at 12 hr to return to basal values after 24 hr. Inhibition of REST activity by ODN delayed the *Npas4* rise and suppressed the return of *Npas4* to basal levels (*Figure 8B*). These results demonstrate that REST affects *Npas4* transcription kinetics in a fashion similar to that of BDNF (see *Figure 6B*) but developing on a shorter time scale. Moreover, the data underline a dual role of REST in (i) the early induction of *Npas4* mRNA and (ii) the late return of *Npas4* mRNA to basal levels. Indeed, when REST is blocked by ODN, the increase of *Npas4* mRNA is delayed and its recovery to control value after 24 hr is inhibited.

4AP-induced hyperactivity also promoted increases in *cds-Bdnf* and *Syt4* mRNAs that were delayed (6 hr) with respect to the raise of *Rest* and *Npas4* transcripts (1 hr); this delayed rise in *cds-Bdnf* and *Syt4* mRNAs was also attenuated by REST inhibition with ODN treatment (*Figure 8C and D*).

Since BDNF is one of the main actors in the development of GABAergic inputs, the fine spatial and temporal regulation of the expression of distinct BDNF-transcripts could contribute to the REST-dependent potentiation of perisomatic GABAergic inhibition. This hypothesis brought us to investigate the mRNAs changes of *P1-*, *P2-*, *P3-*, and *P4-Bdnf* splice variants, known for their high expression in the postnatal hippocampus and cortex and for their activity-dependent regulation (*Aid et al., 2007*) in cultured neurons treated with NEG/vehicle, NEG/4AP, ODN/vehicle, and ODN/4AP for different times. While all *Bdnf* transcripts were increased by hyperactivity (*Figure 8E–H*), only the *P1-Bdnf* mRNA was significantly inhibited by ODN (*Figure 8E*). This specific effect is highly suggestive, in view of the previously demonstrated crucial role that NPAS4 selectively exerts on the activity-dependent transcription of *P1-Bdnf* (*Lin et al., 2008*; *Pruunsild et al., 2011*).

To investigate at the protein level the outcome of the REST-dependent transcriptional changes, cultured neurons (17 div) treated with NEG/vehicle, NEG/4AP, ODN/vehicle, and ODN/4AP for 24 hr were processed by western blotting (*Figure 8I*). The data confirm the previously observed increase of REST expression in 4AP treated neurons (*Pecoraro-Bisogni et al., 2018*; *Pozzi et al., 2013*). Moreover, the increase in REST expression in neurons treated with ODN in the absence of 4AP reveals a negative feedback regulation exerted by REST on its own expression. Twenty-four hours of hyperactivity induced a parallel increase in both NPAS4 and SYT4 protein levels. When 4AP was applied in the presence of ODN, however, the increase of NPAS4 protein was fully suppressed, while SYT4 overexpression was unaffected (*Figure 8J*).

The data confirm the existence of a sequential mechanism for the target-specific enhancement of inhibitory transmission induced by hyperactivity in which REST induces an early activation of the fast immediate early gene NPAS4 that is followed by transcription of BDNF controlled by the P1 promoter (*Aid et al., 2007*).

## Translational correlates of the REST-dependent homeostatic response to hyperactivity

To complete the picture, we analyzed the changes in the transcripts encoding for presynaptic and postsynaptic molecular actors, crucial for structural and functional plasticity of GABAergic synapses, in the same timescale adopted for the transcriptional profiles of *Rest, Npas4, and Bdnf*.

While *Gad2*(GAD65) mRNA was not modulated by hyperactivity, both *Gad1*(GAD67) and *Slc32a1*(VGAT) mRNAs showed significant increases under 4AP stimulation that were fully blocked by ODN treatment (*Figure 9A–C*). Notably, the mRNA peaks of these presynaptic target genes were temporally shifted to 12 hr with respect to *Rest, Npas4,* and *Bdnf* mRNA that peaked within 6 hr from the onset of hyperactivity (see *Figure 8A–H*). These results were confirmed by immunoblot analysis (*Figure 9D*), which revealed a further delayed increase of GAD67 and VGAT protein levels 48 hr after the onset of hyperactivity, that was also suppressed by ODN treatment (*Figure 9E*). The selective increase in GAD67, but not in GAD65 preferentially localized to synaptic terminals, confirms a key role of GAD67, also found at synapses (*Esclapez et al., 1994*; *Fukuda et al., 1998*), in GABA synthesis (*Asada et al., 1997*) and its modulation by neuronal activity and BDNF (*Esclapez and Houser, 1999*; *Hartman et al., 2006*; *Lau and Murthy, 2012*; *Ramírez and Gutiérrez, 2001*; *Rutherford et al., 1997*; *Swanwick et al., 2006*).

The mRNA levels of *Gabra2 and Gabrg2,* and the corresponding protein levels of the postsynaptic GABA$_A$ receptor subunits α$_2$ and γ$_2$, were not affected by hyperactivity (*Figure 9—figure supplement 1*). This lack of effect can be explained by the opposite actions of hyperactivity on the density of

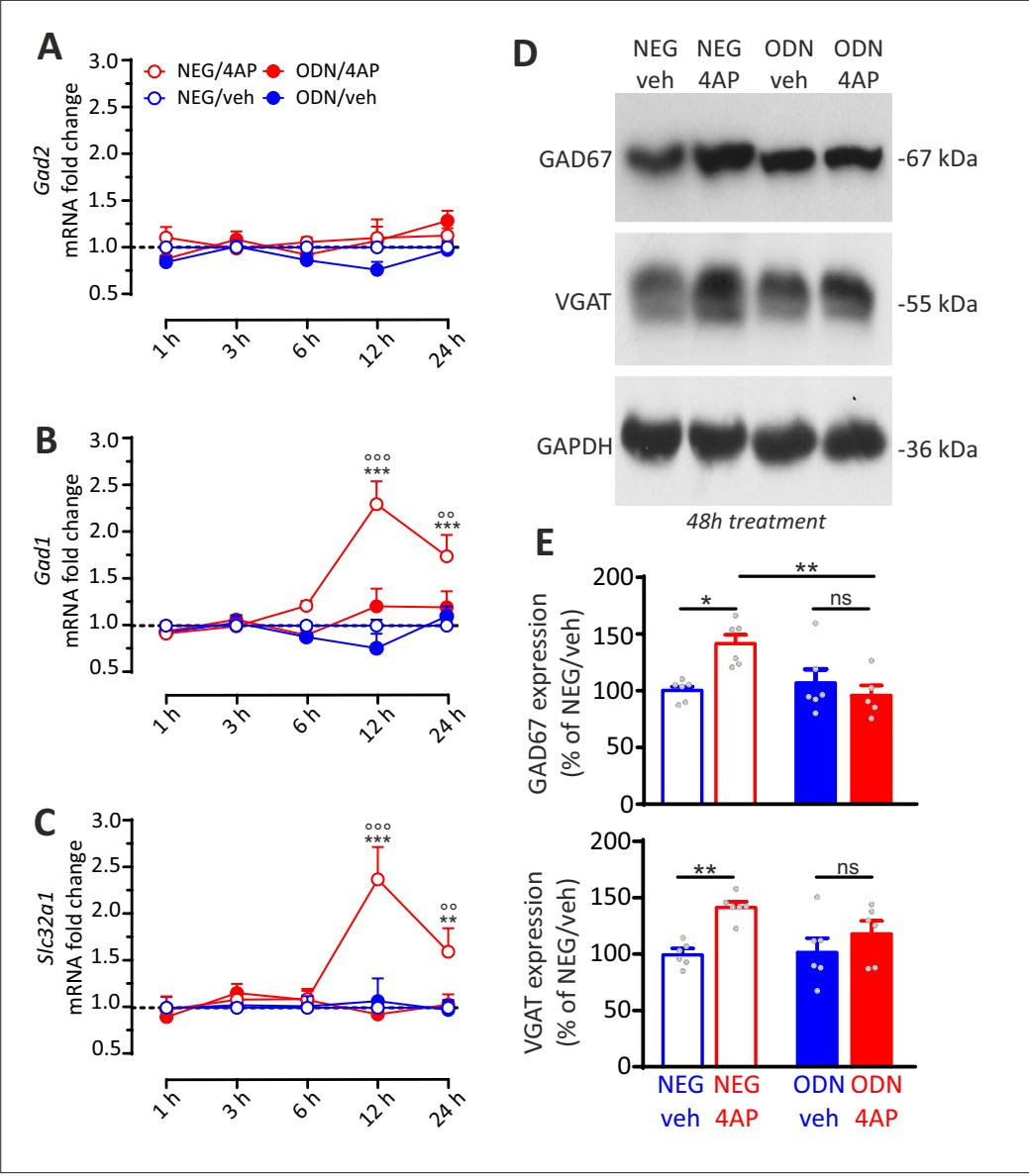

**Figure 9.** REST-dependent activation of GABAergic synaptic genes. (**A–C**) Time course of the fold changes in mRNA levels of *Gad2* (**A**), *Gad1* (**B**), and *Slc32a1* (**C**) in cortical neurons treated with NEG/vehicle, NEG/4AP, ODN/vehicle, or ODN/4AP before and various times after the respective treatments. All values (means± SEM) are normalized to the NEG/veh levels. For each time point, $9 < n < 6$ from three independent neuronal preparations. **p<0.01, ***p<0.001 NEG/veh versus NEG/4AP; °°p<0.01, °°°p<0.001 NEG/4AP versus ODN/4AP; two-way ANOVA/ Tukey's tests. (**D, E**) Representative immunoblots (**D**) and quantitative analysis (**E**) of GAD67 and VGAT protein expression in cortical neurons treated for 48 hr with NEG/vehicle, NEG/4AP, ODN/vehicle, or ODN/4AP. All values (means± SEM) are normalized to the NEG/vehicle level. GAPDH was included as control of equal loading. For each protein, $6 < n < 5$ from three independent neuronal preparations. *p<0.05, **p<0.01; two-way ANOVA/ Tukey's tests. Data for (**A–E**) can be found in *Figure 9—source data 1*.

The online version of this article includes the following figure supplement(s) for figure 9:

**Source data 1.** Source data for *Figure 9*.

**Figure supplement 1.** The levels of α2/ γ 2, GABA$_A$ receptor subunits were not affected by hyperactivity.

**Figure supplement 1—source data 1.** Source data for *Figure 9—figure supplement 1*.

perisomatic and dendritic synapses. On the contrary, the increase in the expression of presynaptic markers could be due to the fact that perisomatic terminals are more active and larger than dendritic ones (*Harney and Jones, 2002*; *Miles et al., 1996*) and that the functional upscaling of perisomatic contacts occurs at the presynaptic level. This can explain why the increase of perisomatic terminals could strongly boost the expression of presynaptic proteins in a way that is not fully compensated by the observed decrease in dendritic synapses.

## Discussion

The dissection of the complex and articulated epigenetic mechanisms that orchestrate synaptic plasticity and excitability of individual excitatory and inhibitory neurons to maintain stability in neural circuits is an intriguing challenge in neuroscience. We previously showed that sustained neuronal hyperactivity causes REST upregulation that is crucial for maintaining neural network homeostasis via downscaling of intrinsic excitability of excitatory neurons and excitatory synaptic transmission (*Pecoraro-Bisogni et al., 2018*; *Pozzi et al., 2013*).

In this paper, we investigated the homeostatic role of REST in the hyperactivity-induced changes of GABAergic transmission by using the experimental paradigm of long-term treatment with the convulsive agent 4AP and applying sequestration of endogenously expressed REST with ODN. In contrast to the previously used RNA interference, that inhibits REST activity by slowly reducing its expression, ODN operates a competitive inhibition of both constitutively and newly expressed REST, allowing a better temporal dissection of the effects of REST activation by hyperactivity (*McClelland et al., 2014*; *McClelland et al., 2011*). A fluorescently tagged variant of ODN demonstrated that REST translocation to the nucleus occurs in both excitatory and inhibitory neurons in response to sustained hyperactivity. However, the resulting responses at inhibitory synapses were opposite to those elicited at excitatory synapses and strikingly target specific: inhibitory synapses on inhibitory neurons were not affected, while inhibitory synapses on excitatory neurons displayed an upscaling of perisomatic inhibition with increased (i) frequency of mIPSCs, (ii) density of perisomatic inhibitory synapses, (iii) amplitude of eIPSCs, and (iv) RRP size, release probability, and transcription of the presynaptic markers GAD67 and VGAT. This potentiation of perisomatic synapses was accompanied by an opposite, milder decrease of the density of dendritic synapses.

The clear-cut target selectivity of these effects suggests that the REST-dependent transcriptional cascades activated by hyperactivity involve a retrograde crosstalk between excitatory neurons and inhibitory terminals, restricting the synaptic changes only to inhibitory→excitatory synapses. This possibility prompted us to investigate the interplay between REST and BDNF. In fact, BDNF is well known for its capability to strengthen GABAergic inputs (*Huang et al., 1999*; *Marty et al., 2000*; *Seil and Drake-Baumann, 2000*) by acting at the presynaptic level (*Baldelli et al., 2005*; *Baldelli et al., 2002*; *Valente et al., 2012*). Moreover, neuronal hyperactivity is known to enhance BDNF transcription and release by excitatory, but not inhibitory, neurons (*Spiegel et al., 2014*; *Hofer et al., 1990*; *Isackson et al., 1991*; *Cohen-Cory et al., 2010*; *Andreska et al., 2014*). Using BDNF scavenging, we demonstrate that BDNF is necessary for the hyperactivity-induced, REST-dependent enhancement of perisomatic inhibitory synapses, while it is not involved in the decrease of dendritic synapses. These results are reminiscent of previous reports showing a BDNF-dependent increase of perisomatic synapses and a BDNF-independent decrease of dendritic synapses onto hippocampal CA1 pyramidal neurons of mice exposed to enriched environment or treated with kainic acid (*Bloodgood et al., 2013*). Moreover, in vivo studies showed that mutant mice lacking activity-dependent P4-BDNF expression exhibit reduced GABA release from parvalbumin-positive fast-spiking interneurons mediating perisomatic inhibition, but not from other interneuronal subtypes (*Jiao et al., 2011*).

From these considerations, BDNF appears as the best candidate downstream effector of the homeostatic effects of REST on perisomatic inhibitory synapses. Indeed: (i) BDNF-TrkB binding inhibition fully recapitulates the effects of REST inhibition on strength/density of perisomatic inhibitory synapses onto excitatory neurons; (ii) REST inhibition dampens the hyperactivity-induced increase in *cds-Bdnf* transcript and, more specifically, of the *P1-Bdnf* transcript; (iii) a sequential temporal profile of transcriptional changes exists with a prompt (1 hr) elevation of *Rest* expression, followed by activation of *cds-Bdnf/P1-Bdnf* transcription (6 hr) and by the late increase in transcription of the presynaptic GABAergic markers *Gad1* and *Slc32a1* (12 hr). In addition, using a 2×2 factorial paradigm combining REST inhibition with the inhibition or stimulation of the TrkB pathway, we demonstrate that REST and

BDNF act *'in series'* and do not simply converge as independent cascades to elicit the changes in peri-somatic inhibitory synapses. In fact, the combined blockade of REST and BDNF activities exerts the same extent of inhibition of the homeostatic response of the respective single treatments (occlusion). On the other hand, exogenous BDNF fully rescues the homeostatic response in the presence of REST inhibition, indicating that it lies downstream of REST activation.

Interestingly, REST had a biphasic effect on BDNF expression in response to hyperactivity, with an early enhancement followed by a late inhibition that confined the BDNF response within a time window of 48 hr. While the latter action is coherent with the transcriptional repressive action of REST on BDNF expression (*Bruce et al., 2004*; *Garriga-Canut et al., 2006*; *Hara et al., 2009*; *Otto et al., 2007*; *Paonessa et al., 2016*), the early enhancement of the BDNF response is somewhat unexpected and may involve an indirect effect. In this respect, it has been shown that the occupancy of RE-1 sites on the promoters of *Rest* target genes can occasionally induce transcriptional activation (*Kallunki et al., 1998*; *Perera et al., 2015*) by favoring the expression of other transcription factors. Relevant to this study, it has been reported that the deletion of the RE-1 element in the *Npas4* promoter exerted a negative effect on *Npas4* transcription (*Bersten et al., 2014*). NPAS4 is one of the most recently identified IEGs with peculiar characteristics (*Lin et al., 2008*; *Sun and Lin, 2016*). It is among the most rapidly induced IEGs, it is neuron-specific and selectively activated by neuronal activity (*Sun and Lin, 2016*). Moreover, NPAS4 is known to upregulate perisomatic inhibitory synapses to excitatory neurons, an effect mediated by the activity-dependent transcription of BDNF (*Bloodgood et al., 2013*; *Spiegel et al., 2014*; *Lin et al., 2008*) that mimics the homeostatic effects of REST described here. This may involve NPAS4 in the transcriptional program initiated by REST and actuated by BDNF.

The analysis of the temporal profiles of *Rest, Npas4, Bdnf,* and *Syt4* transcriptional responses to hyperactivity suggests the existence of a sequential activation of interrelated transcriptional regulators. In fact, both *Rest* and *Npas4* mRNAs increase after 1 hr of heightened activity, while *cds-Bdnf* and *Syt4* mRNAs display a later increase (6 hr) that was partially REST-dependent. It is known that the various *Bdnf* promoters are differentially responsive to neuronal activity (*Aid et al., 2007*). While the *P4* transcript responded more quickly, the *P1* transcript, which showed the highest upregulation, was the only REST-dependent one. This result further supports a REST-NPAS4 link, since NPAS4 is known to specifically control the activity-dependent activation of *P1-Bdnf* transcription (*Pruunsild et al., 2011*). In addition, REST has also a crucial role in the temporal confinement of the transcriptional response to hyperactivity, limiting the *Npas4* and *Bdnf* activation window in time.

It is tempting to speculate that REST by translocating from the cytoplasm to the nucleus, triggers the activation *'Npas4/P1-Bdnf'* gene transcription cascade for the homeostatic rearrangement of perisomatic inhibitory inputs to excitatory neurons (*Figure 10*). An initial transcriptional wave is characterized by the activation of fast IEGs such as *Rest* and *Npas4*, in which the nuclear translocation of REST has an early facilitating role on *Npas4*. A second transcriptional wave involves the expression of the *P1-Bdnf* transcript by NPAS4, which potentially restricts BDNF action to the soma of excitatory neurons (*Brigadski et al., 2005*; *Dean et al., 2012*; *Pattabiraman et al., 2005*). In turn BDNF, secreted by the soma of excitatory neurons, retrogradely reaches the inhibitory presynaptic terminals where it activates, through the TrkB signaling pathway, the final transcriptional wave of effector genes that actuates the homeostatic synaptic changes. The downscaling of dendritic inhibitory synapses to excitatory neurons, occurring in parallel with the homeostatic potentiation of perisomatic synapses, is exclusively dependent on REST, consistent with its well-known transcriptional repressor activity (*Figure 10*). Interestingly, similar opposite changes in the perisomatic and dendritic inhibitory synapses on hippocampal pyramidal neurons were found in experimental models of temporal lobe epilepsy (*Cossart et al., 2001*).

Network electrophysiology with MEAs testified that the complex REST-dependent rearrangement of GABAergic inhibition onto principal neurons is a fundamental component of the homeostatic recovery of the network from hyperactivity. In fact, blockade of GABA$_A$ receptors uncovered an increased synaptic inhibition in response to hyperactivity that was blocked by REST inhibition. A key point is how the REST-mediated changes in response to hyperexcitability mechanistically result in the observed network-wide behavior. Chronic exposure to 4AP activates a complex of REST-dependent changes at the cellular level: (i) reduced expression of voltage-gated Na$_V$1.2 channels, downregulating intrinsic excitability in principal neurons (*Pozzi et al., 2013*); (ii) reduced strength of excitatory synapses to excitatory neurons, acting at the presynaptic level (*Pecoraro-Bisogni et al.,*

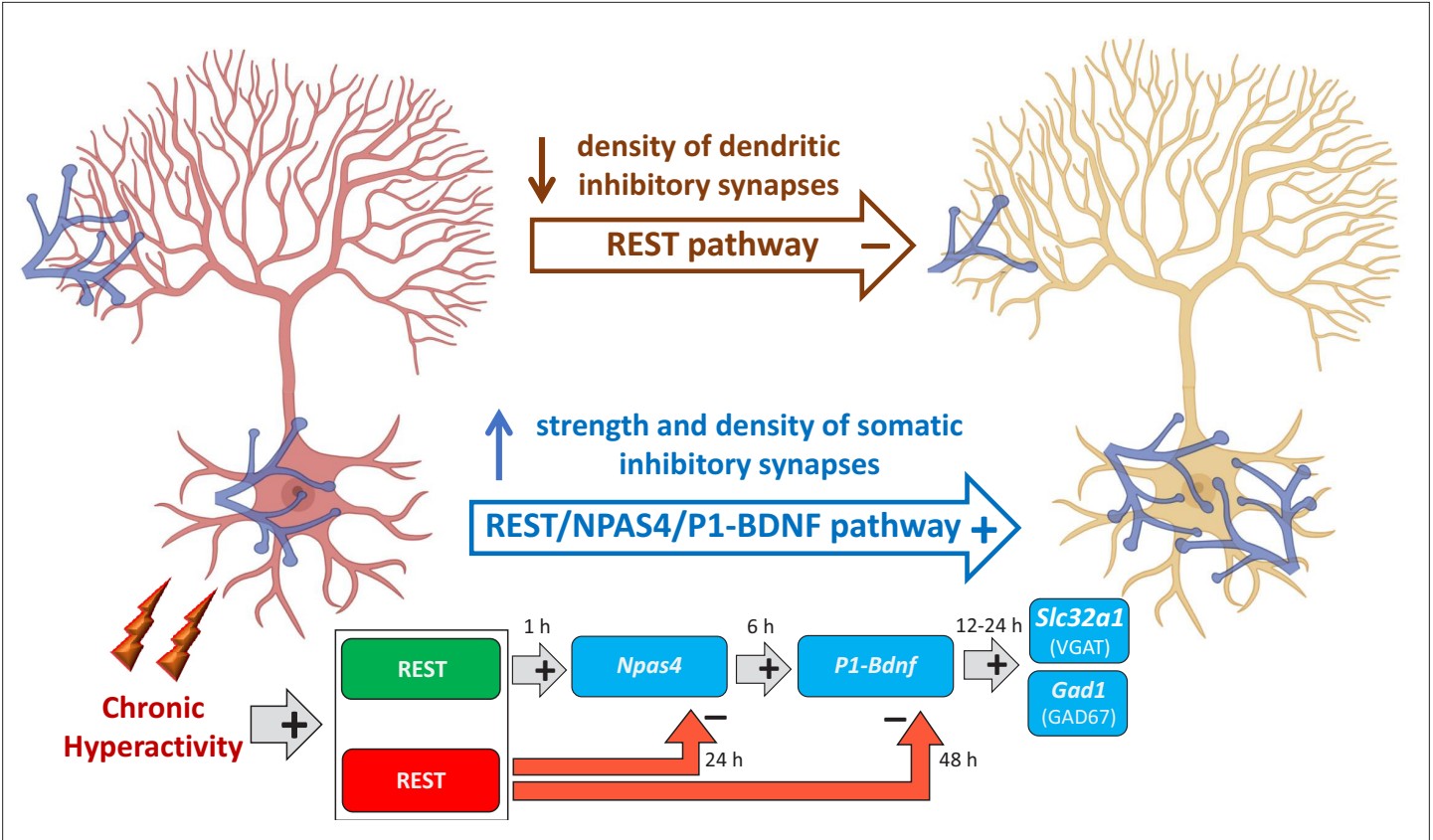

**Figure 10.** Mechanistic model of REST-dependent homeostatic rearrangement of inhibitory inputs in response to hyperactivity. The homeostatic response to hyperactivity at inhibitory synapses onto excitatory principal neurons (in red at the onset of hyperactivity; in yellow after the homeostatic response) consists of the upscaling of perisomatic synapses and a parallel downscaling of dendritic inhibitory synapses. The decrease in dendritic inhibitory synapses is exclusively dependent on REST and independent on BDNF and consists in the canonical effect of transcriptional repression of REST rapidly activated by hyperactivity. The upscaling of perisomatic inhibitory synapses is based on the activation of three sequential transcriptional waves. The first one consists the initial nuclear translocation of REST that exerts an early facilitating role on the fast onset of *Npas4* transcription. The second wave involves the expression of the *P1-Bdnf* transcript by *Npas4* that potentially restricts the functional changes to the soma of excitatory neurons. Finally, in the third wave, BDNF secreted by the excitatory neurons retrogradely reaches the inhibitory terminals and activates, via the TrkB signaling pathway, the transcription of effectors genes (*Slc32a1* and *Gad1*) whose products (VGAT and GAD67) eventually increase the strength of perisomatic inhibitory inputs to excitatory target neurons through a presynaptic mechanism. Notably, REST is also involved in the late transcriptional inhibition of *Npas4* and *Bdnf*, crucial to ensuring a temporal constrain to their expressional increase (created in BioRender.com).

*2018*); (iii) increased strength and density of perisomatic GABAergic synaptic contacts to excitatory neurons; and (iv) decreased density of dendritic inhibitory synapses onto excitatory neurons (this paper). Collectively, the first three effects highlight a dramatic alteration of the E/I balance that can explain the robust homeostatic control of network dynamics characterized by compensation of the BF and moderate overcompensation of MFR. While the enhancement of perisomatic inhibitory inputs represents the main limitation to the repetitive firing of excitatory neurons, the decrease of dendritic inhibition, that normally controls the efficacy of afferent inputs by suppressing dendritic $Ca^{2+}$ spikes and limiting EPSC-induced depolarization (*Miles et al., 1996*), could favor postsynaptic integration and plasticity at the dendritic level.

In conclusion, the results strengthen the idea that the complex epigenetic pathways mediated by REST and downstream transcriptional cascades activate a multiplicity of homeostatic processes acting at different levels. It was previously shown that network stability and compensatory network responses are greater at the population level compared to the single-unit level (*Slomowitz et al., 2015*), indicating that similar network properties may arise from multiple configurations of individual components (*Marder and Goaillard, 2006*; *Prinz et al., 2004*). Consistent with this concept, our results suggest that firing macro-stability can emerge from the distinct combination of changes in

synaptic strength and intrinsic neuronal properties of excitatory and inhibitory neurons operated by the homeostatic cascades initiated by REST.

# Materials and methods

## Key resources table

| Reagent type (species) or resource | Designation | Source or reference | Identifiers | Additional information |
|---|---|---|---|---|
| Genetic reagent (*Mus musculus*) | C57BL6/J | Charles River | | |
| Genetic reagent (*Mus musculus*) | *Gad1*-GFP | *Tamamaki et al., 2003* | | |
| Antibody | beta3-Tubulin (Chicken Polyclonal) | Synaptic Systems | Cat. #302306; RRID:AB_2620048 | IF (1:500) |
| Antibody | VGAT cytoplasmic domain (Mouse Monoclonal) | Synaptic Systems | Cat. #131011; RRID:AB_887872 | IF (1:500) |
| Antibody | Gephyrin (Mouse Monoclonal) | Synaptic Systems | Cat. #147011; RRID:AB_887717 | IF (1:200) |
| Antibody | Alexa405 goat anti-chicken | Abcam | Cat. #ab175674; RRID:AB_2890171 | IF (1:500) |
| Antibody | Alexa546 goat anti-rabbit | Thermo Fisher Scientific | Cat. #A11010; RRID:AB_2534077 | IF (1:500) |
| Antibody | Alexa647 goat anti-mouse | Thermo Fisher Scientific | Cat. #A32728; RRID:AB_2633277 | IF (1:500) |
| Antibody | Anti-REST (Rabbit Polyclonal) | Millipore | Cat. #07-579; RRID:AB_11211936 | WB (1:500) |
| Antibody | NPAS4 (Mouse Monoclonal) | Thermo Fisher Scientific | Cat. #S408-79; RRID: AB_2735296 | WB (1:500) |
| Antibody | Synaptotagmin 4 (Rabbit Polyclonal) | Synaptic Systems | Cat. #105043; RRID:AB_887837 | WB (1:2000) |
| Antibody | Anti-GAD67 (Mouse Monoclonal) | Sigma-Aldrich | Cat. #MAB5406; RRID:AB_2278725 | WB (1:2000) |
| Antibody | Anti-GAPDH (Rabbit Polyclonal) | Santa Cruz | Cat. #Sc-25778; RRID:AB_10167668 | WB (1:2000) |
| Antibody | Anti-mouse Peroxidase conjugated | Bio-Rad | Cat. #1706516; RRID:AB_11125547 | WB (1:3000) |
| Antibody | Anti-rabbit Peroxidase conjugated | Bio-Rad | Cat. #1706515; RRID:AB_11125142 | WB (1:3000) |
| Antibody | GABA-A receptor alpha2 (Rabbit Polyclonal) | Synaptic Systems | Cat. #224 103; RRID:AB_2108839 | WB (1:1000) |
| Antibody | GABA-A receptor gamma2 (Rabbit Polyclonal) | Synaptic Systems | Cat. #224 003; RRID:AB_2263066 | WB (1:1000) |
| Recombinant DNA reagent vectors | pcDNA6.2-GW/EmGFP-miR | Invitrogen | Cat. #K493600 | BLOCK-iT Pol II miR RNAi Expression Vector Kit |
| Recombinant DNA reagent | pCCL.sin.cPPT.PGK.GFP.WPRE | TIGET, San Raffaele Sci. Institute | Gift from M. Amendola and L. Naldini | |
| Sequence-based reagent | ODN (Positive decoy, top) | Sigma Genosys (*Soldati et al., 2011*) | | 5'-GpPpCpPTPTT CAGCACCACGG ACAGCGCCAGC-3' |

*Continued on next page*

*Continued*

| Reagent type (species) or resource | Designation | Source or reference | Identifiers | Additional information |
|---|---|---|---|---|
| Sequence-based reagent | ODN (Positive decoy, bot) | Sigma Genosys (*Soldati et al., 2011*) | | 3'-GpPpCpPTPGGC GCTGTCCGTGG TGCTGAAAGC-5' |
| Sequence-based reagent | NEG (Negative decoy, top) | Sigma Genosys (*Soldati et al., 2011*) | | 5'-GpPpCpPTPTCC AGCACAGTGG TCAGACCC-3' |
| Sequence-based reagent | NEG (Negative decoy, bot) | Sigma Genosys (*Soldati et al., 2011*) | | 3'-GpPpCpPTPTC TGACCACTGTG CTGGAAGC-5' |
| Sequence-based reagent | *Gapdh_F* | This paper | PCR primers | GAACATCATC CCTGCATCCA |
| Sequence-based reagent | *Gapdh_R* | This paper | PCR primers | CCAGTGAGC TTCCCGTTCA |
| Sequence-based reagent | *Ppia_F* | This paper | PCR primers | CACTGTCGC TTTTCGC CGCTTG |
| Sequence-based reagent | *Ppia_R* | This paper | PCR primers | TTTCTGCTGTC TTTGGAACT TTGTCTGC |
| Sequence-based reagent | *Rest_F* | This paper | PCR primers | GAACCACCT CCCAGTATG |
| Sequence-based reagent | *Rest_R* | This paper | PCR primers | CTTCTGACA ATCCTCCATAG |
| Sequence-based reagent | *Bdnf cds_F* | This paper | PCR primers | GATGCCGCAA ACATGTCTATGA |
| Sequence-based reagent | *Bdnf cds_R* | This paper | PCR primers | TAATACTGTCAC ACACGCT CAGCTC |
| Sequence-based reagent | *Bdnf (P1)_F* | This paper | PCR primers | TGGTAACCT CGCTCATT CATTAGA |
| Sequence-based reagent | *Bdnf (P1)_R* | This paper | PCR primers | CCCTTCGCAAT ATCCGCAAAG |
| Sequence-based reagent | *Bdnf (P4)_F* | This paper | PCR primers | CAAATGGAGC TTCTCGC TGAAGGC |
| Sequence-based reagent | *Bdnf (P4)_R* | This paper | PCR primers | GTGGAAATTG CATGGCG GAGGTAA |
| Sequence-based reagent | *Npas4_F* | This paper | PCR primers | AGGGTTTGCT GATGAGTTGC |
| Sequence-based reagent | *Npas4_R* | This paper | PCR primers | CCCCTCCAC TTCCATCTTC |
| Sequence-based reagent | *Gabrg2_F* | This paper | PCR primers | CCATGCCCA ATCCGTGGTTAT |
| Sequence-based reagent | *Gabrg2_R* | This paper | PCR primers | GCTCAAAGG TATCTTGC TCAGTGT |
| Sequence-based reagent | *Gabra2_F* | This paper | PCR primers | GCTGCTCTC CATTGTGAAT |
| Sequence-based reagent | *Gabra2_R* | This paper | PCR primers | GGTGCTGGG AACTTGAAAT |
| Sequence-based reagent | *Gad1_F* | This paper | PCR primers | CTAGGGACC CAGGGAAAG |

*Continued on next page*

*Continued*

| Reagent type (species) or resource | Designation | Source or reference | Identifiers | Additional information |
|---|---|---|---|---|
| Sequence-based reagent | *Gad1_R* | This paper | PCR primers | GTACATCTGTC ATCCATCATCC |
| Sequence-based reagent | *Gad2_F* | This paper | PCR primers | ACCAATTATGG AGCGTCACAGG |
| Sequence-based reagent | *Gad2_R* | This paper | PCR primers | CTGAGGAGCA GCACCTTCTC |
| Sequence-based reagent | *Slc32a1_F* | This paper | PCR primers | TTCAGTGCTT GGAATCTAC |
| Sequence-based reagent | *Slc32a1_R* | This paper | PCR primers | TTCTCCAGAG TGAAGTCG |
| Sequence-based reagent | *Syt4_F* | This paper | PCR primers | CCTCACTCAT CGCCATCCA |
| Sequence-based reagent | *Syt4_R* | This paper | PCR primers | GACCGCAGC TCACTCCAT |
| Chemical compound, drug | Trypsin | Gibco | Cat. #1505014 | (0.25%) |
| Chemical compound, drug | HBSS | Gibco | Cat. #14170-088 | |
| Chemical compound, drug | HEPES | AppliChem | Cat. #A37240500 | (10 mM) |
| Chemical compound, drug | D-Glucose | Sigma-Aldrich | Cat. #G7021 | (30 mM) |
| Chemical compound, drug | Gentamicin solution | Sigma-Aldrich | Cat. #G1272 | (5 µg/ml) |
| Chemical compound, drug | KOH | AppliChem | Cat. #131515 | (1 M) |
| Chemical compound, drug | Bovine Serum Albumin (BSA) | Sigma-Aldrich | Cat. #A4503 | (10%) |
| Chemical compound, drug | $MgSO_4 + H_2O$ | Sigma-Aldrich | Cat. #63138 | (6 mM) |
| Chemical compound, drug | Poly-L-lysine hydrobromide | Sigma-Aldrich | Cat. #P2636 | (0.1 mg/ml) |
| Chemical compound, drug | Boric Acid | Sigma-Aldrich | Cat. #B-7901 | (0.1 M) |
| Chemical compound, drug | di-Sodium tetraborate decahydrate | Sigma-Aldrich | Cat. #A267508 | (0.1 M) |
| Chemical compound, drug | Neurobasal-A Medium | Gibco | Cat. #10888-022 | |
| Chemical compound, drug | Neurobasal Medium | Gibco | Cat. #21103049 | |
| Chemical compound, drug | B-27 Supplement | Gibco | Cat. #17504044 | (2%) |
| Chemical compound, drug | GlutaMAX -I (100×) | Gibco | Cat. #35050-061 | (1 mM) |
| Chemical compound, drug | Penicillin-Streptomycin | Sigma-Aldrich | Cat. #P4333 | (1%) |
| Chemical compound, drug | 4-Aminopyridine | Sigma-Aldrich | Cat. #A78403 | (100 µM) |
| Chemical compound, drug | $CaCl_2$ | Sigma-Aldrich | Cat. #C8106 | (2 mM) |
| Chemical compound, drug | $MgCl_2$ | Sigma-Aldrich | Cat. #A4425 | (1 mM) |
| Chemical compound, drug | KCl | Merk | Cat. #TA638736 | (4 mM) |
| Chemical compound, drug | ATP | Sigma-Aldrich | Cat. #A6419 | (3 mM) |
| Chemical compound, drug | GTP | Sigma-Aldrich | Cat. #G8877 | (0.1 mM) |
| Chemical compound, drug | EGTA | Sigma-Aldrich | Cat. #E4378 | (0.1 mM) |
| Chemical compound, drug | TrkB-Fc | Sigma-Aldrich | Cat. #T8694 | (1 µg/ml) |

*Continued*

| Reagent type (species) or resource | Designation | Source or reference | Identifiers | Additional information |
|---|---|---|---|---|
| Chemical compound, drug | Triton X-100 | Sigma-Aldrich | Cat. #T9284 | (0,2%) |
| Chemical compound, drug | Sucrose | AppliChem | Cat. #A2211 | (4%) |
| Chemical compound, drug | Phosphate-buffered saline | Sigma-Aldrich | Cat. #P4417 | (1 ×) |
| Chemical compound, drug | Paraformaldehyde | Sigma-Aldrich | Cat. #P6148 | (4%) |
| Chemical compound, drug | Fetal Bovine Serum, qualified, heat | Gibco | Cat. #10500064 | (5%) |
| Chemical compound, drug | Tetrodotoxin (TTX) | Tocris | Cat. #1078 | (1 µM) |
| Chemical compound, drug | D-AP5 | Tocris | Cat. #0106 | (50 µM) |
| Chemical compound, drug | CNQX | Tocris | Cat. #0190 | (10 µM) |
| Chemical compound, drug | QX314 | Tocris | Cat. #1014 | (10 mM) |
| Chemical compound, drug | EDTA | AppliChem | Cat. #A5097 | (10 mM) |
| Chemical compound, drug | NaCl | Sigma-Aldrich | Cat. #793566 | (1 M) |
| Chemical compound, drug | Tris-HCl | Merk | Cat. #10812846001 | (100 mM) |
| Chemical compound, drug | Tris(2-carboxyethyl) phosphine (TCEP) | Sigma-Aldrich | Cat. #C4706 | (50 mM) |
| Chemical compound, drug | Protease inhibitor cocktail | Cell Signaling | Cat. #5871 | |
| Chemical compound, drug | BCA Assay | Thermo Fisher Scientific | Cat. #23225 | |
| Chemical compound, drug | Hoechst 33342 | Thermo Fisher Scientific | Cat. #H1399 | (3.33 µl/ml) |
| Chemical compound, drug | TRIzol Reagent | Invitrogen | Cat. #15596018 | |
| Chemical compound, drug | Brain-derived neurotrophic factor (BDNF) | Sigma-Aldrich | Cat. #B3795 | (100 ng/ml) |
| Software, algorithm | GraphPad Prism | GraphPad Software, Inc | RRID:SCR_002798 | |
| Software, algorithm | PatchMaster | HEKA Inst | RRID:SCR_000034 | |
| Software, algorithm | Fitmaster | HEKA Inst | RRID:SCR_016233 | |
| Software, algorithm | ImageJ, Fiji | (ImageJ; http://rsb.info.nih.gov/ij) | RRID:SCR_002285 | |
| Software, algorithm | LASX | Leica Microsystems | RRID:SCR_013673 | |
| Software, algorithm | AxIS 2.4 | Axion BioSystems | RRID:SCR_016308 | |

## Experimental animals

*Gad1*-GFP knock-in mice were generated by inserting the cDNA encoding enhanced GFP into the Gad1 locus in TT2 embryonic stem cells, as described (*Tamamaki et al., 2003*). Heterozygous *Gad1*-GFP males were mated with wild-type C57BL6/J females, and GFP-positive pups were identified at birth through a Dual Fluorescent Protein Flashlight (DFP-1, NIGHTSEA, Lexington, MA) and confirmed by genotyping, performed by PCR with the following primers: TR-1b: GGCACAGCTCTC CCTTCTGTTTGC; TR-3: GCTCTCCTTTCGCGTTCCGACAG; TRGFP-8: CTGCTTGTCGGCCATGATAT AGACG. All animals were provided by our institutional animal breeding facility in accordance with the guidelines approved by the local Animal Care Committee of the University of Genova. All experiments were carried out in accordance with the guidelines established by the European Communities Council (Directive 2010/63/EU of March 4, 2014) and were approved by the Italian Ministry of Health (authorization 73/2014-PR and 1276/2015-PR).

## Cell cultures

Primary hippocampal and cortical neurons were prepared from postnatal *Gad1*-GFP knock-in mice (P0–P1), as previously described (*Beaudoin et al., 2012*; *Prestigio et al., 2019*). Experiments used 0/2-day-old pups. Some control experiments were done using wild-type C57BL6/J mice. Briefly, hippocampi and cortex were dissociated by enzymatic digestion in 0.25% trypsin for 6 min at 37°C and then triturated with a fire-polished Pasteur pipette. No antimitotic drugs were added to prevent glia proliferation. The following solutions were used for cell culture preparations: HANKS solution, prepared from HBSS (GIBCO 14170-088; red) supplemented with 10 mM HEPES, 30 mM D-glucose, 5 µg/ml Gentamycin, pH 7.4 with KOH; dissection solution, prepared from HANKS solution supplemented with 10% bovine serum albumin (BSA) and 6 mM $MgSO_4$. Primary hippocampal neurons were plated at density of 120 cells/mm$^2$ on 3.5 cm diameter Petri dishes (Falcon 35 mm, 353001) treated for 24 hr with poly-L-lysine (0.1 mg/ml; Sigma-Aldrich) in borate buffer (0.1 M). Cells were grown in a culture medium consisting of Neurobasal A (Gibco) supplemented with 2% B-27 (Invitrogen, Italy), 1 mM Glutamax, and 5 µg/ml Gentamycin and maintained at 37°C in a humidified incubator with 5% $CO_2$. To induce neuronal hyperactivity, 4AP, a non-selective K$^+$ channel blocker, was applied on mature neurons, at 17–18 days in vitro (div). 4AP-treatment represents a widely adopted model of drug-induced epileptiform activity in cultured neurons and brain slices (*Avoli and Jefferys, 2016*) that we have previously used in cultured neurons to induce network hyperactivity (*Pecoraro-Bisogni et al., 2018*; *Pozzi et al., 2013*). 4AP (100 µM) was dissolved in complete Neurobasal medium, applied to neuronal cultures, and maintained for the entire indicated times. At this concentration, we have previously demonstrated (*Pozzi et al., 2013*) that the treatment with 4AP for 24, 48 but also 96 hr did not affect viability of cultured neurons.

## Oligodeoxynucleotides decoy

REST activity was blocked by using oligodeoxynuclotides (ODNs) that act as surrogate binding sites for REST and sequester the native transcription factor from its genomic binding sites (*Johnson et al., 2006*; *McClelland et al., 2011*; *Soldati et al., 2011*). The ODN decoy (ODN) was designed corresponding to the canonical REST-binding site, RE1, while a negative decoy control (NEG) was generated using a sequence corresponding to a non-canonical RE1 that does not bind REST (*Bruce et al., 2004*). The decoy ODN sequences were:

> ODN (Positive decoy): (Top) 5′-GpPpCpPTPTTCAGCACCACGGACAGCGCCAGC-3′
> (Bot) 3′-GpPpCpPTPGGCGCTGTCCGTGGTGCTGAAAGC-5′;
> NEG (Negative decoy) (Top) 5′-GpPpCpPTPTCCAGCACAGTGGTCAGACCC-3′
> (Bot) 3′-GpPpCpPTPTCTGACCACTGTGCTGGAAGC-5′.

ODNs were designed with phosphorothioate modification on the first three nucleotides to avoid degradation (*Lee et al., 2003*; *Osako et al., 2007*). Single-stranded oligodeoxynucleotides were synthesized by Sigma Genosys (St. Louis, MO). Annealing was performed in 10× buffer (100 mM Tris-HCl, pH 8.0, 10 mM EDTA, and 1 M NaCl) by heating to at least 5–10°C above their melting temperature and cooling slowly using a heat block. Cultured neurons were treated with 200 nM decoys ODNs. For the live-cell imaging of activity-dependent REST translocation from the cytoplasm to the nucleus, we used the following fluorescent ODNs:

> ODN-Top-Cy3: 5′-GpPpCpPTP TTCAGCACCACGGACAGCGCCAGC-Cy-3;
> NEG-Top-Cy3: 5′-GpPpCpPTPTCCAGCACAGTGGTCAGACCC-Cy3-3′.

## RNA interference, lentivirus production, and infection procedures

The REST target sequence used to design the shRNA that we employed in *Figure 5B* was: 5′-ACAT GCAAGACAGGTTCACAA-3′. This shRNA together with an alternative one were previously validated for their capability of decreasing *Rest* mRNA levels in neuroblastoma and cultured cortical neurons (*Pecoraro-Bisogni et al., 2018*; *Pozzi et al., 2013*). Briefly, the shRNA construct was obtained by cloning the sequence into pcDNA6.2-GW/EmGFP-miR plasmid using a microRNA (miR)-based expression vector kit (BLOCK-iT Pol II miR; Invitrogen), thereby creating an expression cassette consisting of the 5′ miR flanking region, the REST target sequence, and the 3′ miR flanking region. As a negative control, the pcDNA6.2-GW/EmGFP-miR-neg plasmid (Invitrogen), containing a sequence not targeting any known vertebrate gene, was used. The shRNA was then sub-cloned into the lentiviral

vector pCCL.sin.cPPT.PGK.GFP.WPRE (a kind gift from M. Amendola and L. Naldini, TIGET, San Raffaele Sci. Institute, Milan, Italy). Cultured neurons were infected at 14 div by using 10 multiplicity of infection, and neurons were used at 18–20 div.

## Multielectrode array recordings

Neuronal network activity was recorded using a multi-well MEA system (Maestro, Axion BioSystems, Atlanta, GA). The MEA plates (M768-tMEA-48W) were composed of 48 wells, each containing a square grid of 16 PEDOT electrodes (50 µm electrode diameter; 350 µm center-to-center spacing) that creates a 1.1×1.1 mm$^2$ recording area. Spiking activity from networks grown onto MEAs was recorded and monitored using Axion BioSystems hardware (Maestro1 amplifier and Middle-man data acquisition interface) and the Axion's Integrated Studio software in Spontaneous Neural Configuration (AxIS 2.4). After 1200× amplification, raw data were digitized at 12.5 kHz/channel and stored on a hard disk for subsequent offline analysis. The day before culture preparation, MEAs were coated by depositing a 20 µl drop of poly-L-lysine (0.1 mg/ml, Sigma-Aldrich) over each recording area and subsequently incubated overnight. After thorough washing, dissociated cortical neurons were plated at a density of 50,000 per well in a 25 µl drop that covered a surface of 20 mm$^2$, yielding a final cell density of 2500 cells/mm$^2$. Cells were incubated in Neurobasal medium supplemented with 1% Glutamax, 2% B27, and 1% penicillin/streptomycin. Half-volume replacement of the culture medium was performed every 3 days.

Experiments were performed in culture medium maintained at 37°C. Under these culture conditions, we previously observed (*Pozzi et al., 2013*), a time-dependent increase of the activity of the neuronal network, corresponding to the functional maturation of the network with elongation of neural processes and increase of the functional synaptic connectivity.

At div 17, MEA plates were set on the Maestro apparatus and their spontaneous activity recorded for 10 min in standard culture medium (basal condition). Cultured neurons were then treated with either ODN or NEG (200 nM) in the presence of 4AP (100 µM) or vehicle. To remove the effects of mechanical perturbations and to allow neuronal activity to reach a steady state under 4AP, MEA plates were returned to the incubator and recordings were collected 1, 24, and 48 hr after the treatment. On the last day of recording, networks were acutely exposed to the GABA$_A$R blocker bicuculline (BIC, 30 µM, Tocris Bioscience). After 5 min re-equilibration, recordings continued for 10 min (*Valente et al., 2019*). Spike and burst detection were both computed using the Axion BioSystems software NeuralMetricTool. To study firing and bursting properties, only the wells that contained ≥4 active electrodes (≥5 spikes/min) were retained for further analysis. Moreover, we excluded from the analysis the 4AP-untreated wells showing a time-dependent increase of the spontaneous activity higher than 60% during the 48 hr of observation and the 4AP-treated wells that were unresponsive to 4AP. Extracellular APs were detected by adaptive threshold crossing (7× the standard deviation of the rms-noise on each channel) on 200 Hz high-pass-filtered traces. Bursts within single channels were identified by applying an inter-spike interval (ISI) threshold algorithm (*Chiappalone et al., 2006*) that defines bursts as collections of a minimum number of spikes (N$_{min}$=5) separated by a maximum ISI (ISI$_{max}$) of 100 ms. Electrodes that recorded less than five spikes/min were deemed inactive and were not included in the burst analysis.

## Patch-clamp recordings

Whole-cell patch-clamp recordings were performed in the voltage-clamp configuration on hippocampal neurons plated at density of 120 cells/mm$^2$. Electrophysiological experiments were performed at 17–19 div after treatment with either ODN or NEG (200 nM) in the presence of 4AP (100 µM) or vehicle for 48 hr. Patch pipettes, prepared from thin borosilicate glass (Kimble, Kimax, Mexico), were pulled and fire-polished to a final resistance of 3–4 MΩ when filled with the intracellular solution. IPSCs were recorded in Tyrode extracellular solution containing (in mM): 140 NaCl, 2 CaCl$_2$, 1 MgCl$_2$, 4 KCl, 10 glucose, and 10 HEPES (pH 7.3 with NaOH), to which D-2-amino-5-phosphonopentanoic acid (D-AP5; 50 µM), 6-cyano-7-nitroquinoxaline-2,3-dione (CNQX; 10 µM), and N-(2,6-dimethylphenylcarbamoylmethyl)triethylammonium chloride (QX314; 10 mM) were added to block NMDA receptors, non-NMDA receptors, and voltage-activated Na$^+$ channels, respectively. The internal solution composition was (in mM): 140 KCl, 4 NaCl, 1 MgSO$_4$, 0.1 EGTA, 15 glucose, 5 HEPES, 3 ATP, and 0.1 GTP (pH 7.2 with KOH). All recordings were performed at 22–24 °C. Under this

condition, internal and external chloride concentrations were equimolar, shifting the chloride reversal potential from a negative value to 0 mV. This experimental configuration, that transforms IPSCs in inward currents, is typically used for increasing the amplitude of the IPSCs, evoked at negative holding potentials. IPSCs were acquired at 20 kHz sample frequency and filtered at half the acquisition rate with an 8-pole low-pass Bessel filter. Patch-clamp recordings with leak currents >200 pA or series resistance >15 MΩ were discarded. Series resistance was compensated 80% (2 μs response time) and the compensation was readjusted before each stimulation. The shown potentials were not corrected for the measured liquid junction potential (9 mV). Data acquisition was performed using PatchMaster and analyzed with Fit master programs (HEKA Elektronik).

For recording eIPSCs, the postsynaptic *Gad1*-GFP negative excitatory cell was clamped at –70 mV and the soma of the presynaptic *Gad1*-GFP positive interneuron was stimulated through a glass electrode (1 μm tip diameter) filled with Tyrode solution in a *'loose patch configuration.'* The stimulating extracellular pipette delivered biphasic current pulses lasting 0.5 ms of variable amplitude (50–150 mA) by an isolated pulse stimulator (model 2100; A-M Systems, Carlsburg, WA). Monosynaptically connected neurons were identified by the short latency (2–4 ms) necessary to induce eIPSCs. To ensure that only the synaptic contacts of the selected presynaptic neuron were stimulated by the extracellular stimulating pipette, we recorded only those eIPSCs that were completely lost after a few μm displacements from the soma of the presynaptic neuron. Considering that the evoked currents remained stable for stimulation intensities twofold the threshold, the stimulation intensity was set at 1.5-fold the threshold for all experiments. The current artifact produced by the presynaptic extracellular stimulation was subtracted in all the eIPSC traces. Only cells with resting membrane potentials between –57 and –64 mV were considered for analysis. eIPSCs were evoked by two consecutive stimuli separated by a time interval of 50 ms for calculating the paired-pulse ratio (PPR; $I_2/I_1$), where $I_1$ and $I_2$ are the amplitudes of the eIPSCs evoked by the conditioning and test stimuli, respectively. The amplitude of $I_2$ was determined as the difference between the $I_2$ peak and the corresponding value of $I_1$ calculated by mono-exponential fitting of the eIPSC decay. A similar experimental protocol was used to evaluate whether BDNF played a functional role in the 4AP-induced strengthening of inhibitory synapses. To block BDNF activity, 17 div cultured neurons were incubated with TrkB-Fc (1 μg/mL, T8694-Sigma), a BDNF scavenger suppressing BDNF binding to TrkB (*Sakuragi et al., 2013*; *Shelton et al., 1995*).

The size of the RRP of synchronous release ($RRP_{syn}$) and the probability for any given synaptic vesicle in the RRP to be released ($P_r$) were calculated using the cumulative amplitude analysis (*Schneggenburger et al., 1999*). High-frequency stimulation (2 s at 20 Hz) was applied to presynaptic fibers with the extracellular electrode and the $RRP_{syn}$ was determined by summing up peak IPSC amplitudes during the 40 stimuli. The analysis assumes that the depression induced by the train is limited by a constant recycling of synaptic vesicles and that equilibrium is present between released and recycled vesicles. The number of data points to include in the fit of the steady-state phase was evaluated by calculating, for each cell, the best linear fit which included the maximal number of data points starting from the last one. According to this procedure, the intercept with the y-axis gives an estimation of the size of the $RRP_{syn}$ and the ratio between the amplitude of the first eIPSC evoked by the stimulation train ($I_1$) and $RRP_{syn}$ yields an estimation of $P_r$.

mIPSCs were recorded from hippocampal *Gad1*-GFP negative excitatory and *Gad1*-GFP positive inhibitory neurons incubated in the presence of tetrodotoxin (TTX, 1 μM; Tocris) to block spontaneous APs. mIPSC analysis was performed by using the Minianalysis program (Synaptosoft, Leonia, NJ) and the Prism software (GraphPad Software, Inc). The amplitude and frequency of mIPSCs were calculated using a peak detector function with a threshold amplitude set at 4 pA and a threshold area at 50 ms*pA.

## Immunocytochemistry

Immunocytochemistry of primary hippocampal neurons obtained from postnatal *Gad1*-GFP knock-in mice was performed after treatment with either ODN or NEG (200 nM) in the presence of 4AP (100 μM) or vehicle for 48 hr. Neurons were fixed at 19 div with 4% paraformaldehyde/4% sucrose for 12 min at room temperature and then washed with phosphate-buffered saline (PBS). Cells were then permeabilized with methanol (–20°C; 10 min on ice) followed by 0.2% Triton X-100 for 10 min (*Liao et al., 2010*) and washed for 30 min with PBS supplemented with 5% fetal bovine serum (FBS)/0.1%

BSA. Incubation with primary antibodies in PBS/5% FBS/0.1% BSA was performed for 2 hr, followed by washing with PBS and final incubation with secondary antibodies for 2 hr. Neurons were immunostained with antibodies to chicken β3-Tubulin (1:500, Synaptic Systems 302306), rabbit VGAT (1:500, Synaptic Systems 131011) and mouse Gephyrin (1:200, Synaptic Systems 147011). Secondary antibodies were Alexa405 goat anti-chicken (Abcam, ab175674), Alexa546 goat anti-rabbit, and Alexa647 goat anti-mouse (1:500 in all cases, Thermo Fisher Scientific, Cat. A11010, A32728).

Confocal images were acquired by using a 63× oil objective (N.A. 1.4) in a Leica TCS SP8 Confocal Laser Scanning Microscope equipped with Hybrid Detectors (Leica Microsystem). Images were processed and analyzed using the ImageJ software (https://imagej.nih.gov/ij/). For capturing in-focus images of objects at high magnification, multiple (25) images were taken at increasing focal distances (0.3 μm increments) and Z-stacking image processing was used to obtain a composite image with a greater depth of field than individual source images. Analysis of fluorescence intensity was performed on dendritic linear regions of interest (ROIs; 3–5 per image) of 60–160 μm in length and somatic surface ROIs (2–3 per image) of 1000–2000 $\mu m^2$ under blind conditions. The fluorescence intensity profiles were analyzed in each ROI and putative inhibitory synaptic contacts were identified as the ROIs in which both VGAT and gephyrin average intensity profiles exceeded a threshold level (set at three times the background intensity). The synaptic density on dendrites was obtained by counting the total number of positive puncta divided by the length of the tubulin-positive segment (in μm). The synaptic density on the soma was obtained by counting the total number of positive puncta divided by the tubulin-positive soma area (in $\mu m^2$). The indicated sample number (n) represents the number of coverslips collected from at least three independent neuronal preparations. From each coverslip, at least 10–15 images were acquired.

## Live imaging of the REST translocation

Cultured hippocampal neurons (17 div) from *Gad1*-GFP knock-in mice were treated with either vehicle or 4AP for 1 hr and then incubated for 24 hr with either ODN or NEG decoy tagged with Cyanine-3, (Cy3-ODN; Cy3-NEG). Five minutes before images acquisition, neurons were washed with Tyrode solution, stained with Hoechst-333342 (3.33 μl/ml) and further washed two times with Tyrode solution. Differential interference contrast and fluorescence images were acquired using an Olympus IX71 microscope with a 40× objective (Olympus LCPlanFI 40×) equipped with a Hamamatsu (ORCA-ER) camera and a Leica EL6000 fluorescence lamp. Somatic and nuclear Cy3-positive ROIs were drawn for each neuron to calculate the REST partition between the cytosol and the nucleus (Cy3-area$_{nucleus}$/Cy3-area$_{cyto}$ ratio) in stimulated and control excitatory and inhibitory neurons. Images were analyzed using the ImageJ software.

## Real-time qPCR

RNA was extracted with TRIzol reagent and purified on RNeasy spin columns (Qiagen). RNA samples were quantified at 260 nm with an ND1000 Nanodrop spectrophotometer (Thermo Fisher Scientific). RNA purity was also determined by absorbance at 280 and 230 nm. All samples showed A260/280 and A260/230 ratios greater than 1.9. Reverse transcription was performed according to the manufacturer's recommendations on 1 μg of RNA with the QuantiTect Reverse Transcription Kit (Qiagen), which includes a genomic DNA-removal step. SYBR green RT-qPCR was performed in triplicate with 10 ng of template cDNA using QuantiTect Master Mix (Qiagen) on a 7900-HT Fast Real-Time System (Applied Biosystems) as previously described (*Pozzi et al., 2013*), using the following conditions: 5 min at 95°C, 40 cycles of denaturation at 95°C for 15 s, and annealing/extension at 60°C for 30 s. Product specificity and occurrence of primer dimers were verified by melting-curve analysis. Primers were designed with Beacon Designer software (Premier Biosoft) to avoid template secondary structure and significant cross homology with other genes by BLAST search. The PCR reaction efficiency for each primer pair was calculated by the standard curve method with four serial-dilution points of cDNA. The PCR efficiency calculated for each primer set was used for subsequent analysis. All experimental samples were detected within the linear range of the assay. Gene-expression data were normalized via the multiple-internal-control-gene method (*Vandesompele et al., 2002*) with the GeNorm algorithm available in qBasePlus software (Biogazelle).

The control genes used were GAPDH (glyceraldehyde-3-phosphate dehydrogenase) and PPIA (peptidylprolyl isomerase), the expression of these genes was found not to be affected by the 4AP treatment. Primers sequences (5'–3') were indicated in the Key Resources Table.

## Protein extraction and western blotting

Total cell lysates were obtained from 18/19 div cortical neurons treated as previously described. Neurons were lysed in lysis buffer (150 mM NaCl, 50 mM Tris, 1 mM EDTA, and 1% Triton X-100) supplemented with protease inhibitor cocktail (Cell Signaling, Danvers, MA). After 10 min of incubation, lysates were collected and clarified by centrifugation (10 min at 10,000×$g$). Protein concentrations were determined by the BCA protein assay (Thermo Fisher Scientific, Waltham, MA). Sodium dodecyl sulfate polyacrylamide gel electrophoresis (SDS-PAGE) was performed according to Laemmli (1970); equivalent amounts of protein were separated on 8% polyacrylamide gels and transferred onto nitrocellulose membranes (Whatman, St. Louis, MO). Membranes were blocked for 1 hr in 5% non-fat dry milk in Tris-buffered saline (10 mM Tris, 150 mM NaCl, and pH 8.0) plus 0.1% Triton X-100 and incubated overnight at 4°C or for 2 hr at room temperature with the following primary antibodies: anti-REST (1:500; 07-579 Millipore, MA), anti-NPAS4 (1:500; S408-79 Thermo Fisher Scientific, Waltham, MA), anti-SYT4 (1:2000; 105043 Synaptic Systems, Gottingen, Germany), anti-GAD67 (1:2000; MAB5406, Sigma-Aldrich, St. Louis, MO), anti VGAT (1:1000; 131002 Synaptic Systems, Gottingen, Germany), and anti-GAPDH (1:2000; sc-25778 Santa Cruz Biotechnology, Dallas, TX). After several washes, membranes were incubated for 1 hr at room temperature with peroxidase-conjugated anti-mouse (1:3000; Bio-Rad, Hercules, CA) or anti-rabbit (1:3000; Bio-Rad, Hercules, CA) antibodies. Bands were revealed with the ECL chemiluminescence detection system (Thermo Fisher Scientific, Waltham, MA). Immunoblots were quantified by densitometric analysis of the fluorograms (ChemiDoc XRS+; Bio-Rad, Hercules, CA).

## Statistical analysis

Data are given as means± SEM for n=sample size. The normal distribution of experimental data was assessed using Kolmogorov-Smirnov normality test. To compare two normally distributed sample groups, the unpaired two-tailed Student's t-test was used. To compare two sample groups that were not normally distributed, the Mann–Whitney U-test was used. To compare more than two normally distributed sample groups, we used one- or two-way ANOVA, followed by the Tukey's or Dunnett's multiple comparison tests. Alpha levels for all tests were 0.5% (95% confidence intervals). Statistical analysis was carried out by using the SPSS (version 21, IBM Software) and the Prism software (GraphPad Software, Inc).

## Acknowledgements

The authors thank Dr Yuchio Yanagawa (Gunma University, Maebashi, Japan) for kindly providing *Gad1*-GFP knock-in mice; Drs Anna Rocchi and Andrea Contestabile (Istituto Italiano di Tecnologia, Genova, Italy) for help and advice for RT-qPCR experiments; Drs M Amendola and L Naldini (TIGET, San Raffaele Sci. Institute, Milan, Italy) for kindly providing the lentiviral vectors; Drs Silvia Casagrande (Department Experimental Medicine, University of Genova) and Arta Mehilli (Istituto Italiano di Tecnologia, Genova, Italy) for help in the preparation of primary cultures; Drs Riccardo Navone and Diego Moruzzo (Istituto Italiano di Tecnologia, Genova, Italy) for help in the maintenance and genotyping of mouse strains.

## Additional information

### Funding

| Funder | Grant reference number | Author |
| --- | --- | --- |
| Compagnia di San Paolo | 2017.20612 | Pietro Baldelli |
| Compagnia di San Paolo | 2019.34760 | Fabio Benfenati |

| Funder | Grant reference number | Author |
|---|---|---|
| H2020 Health | SNAREopathies | Fabio Benfenati |
| IRCCS Ospedale Policlinico San Martino Genova | Ricerca Corrente and "5x1000 | Pietro Baldelli<br>Fabio Benfenati |
| Ministero dell'Istruzione, dell'Università e della Ricerca | PRIN2017-A9MK4R | Fabio Benfenati |
| Fondazione Telethon | Grant GGP19120 | Fabio Benfenati |
| Italian Ministry of Foreign Affairs and International Cooperation | MAECI EPNZ008201 | Fabio Benfenati |

The funders had no role in study design, data collection and interpretation, or the decision to submit the work for publication.

## Author contributions

Cosimo Prestigio, Formal analysis, Investigation, Methodology, Writing – original draft; Daniele Ferrante, Data curation, Formal analysis, Investigation, Methodology; Antonella Marte, Alessandra Romei, Franco Onofri, Pierluigi Valente, Formal analysis, Investigation, Methodology; Gabriele Lignani, Data curation, Formal analysis, Visualization; Fabio Benfenati, Formal analysis, Funding acquisition, Project administration, Resources, Validation, Visualization, Writing - review and editing; Pietro Baldelli, Conceptualization, Data curation, Formal analysis, Funding acquisition, Investigation, Methodology, Supervision, Validation, Writing – original draft, Writing - review and editing

## Author ORCIDs

Gabriele Lignani (iD) http://orcid.org/0000-0002-3963-9296
Fabio Benfenati (iD) http://orcid.org/0000-0002-0653-8368
Pietro Baldelli (iD) http://orcid.org/0000-0001-9599-3436

## Ethics

All experiments were carried out in accordance with the guidelines established by the European Communities Council (Directive 2010/63/EU of 4 March 2014) and were approved by the Italian Ministry of Health (authorization 73/2014-PR and 1276/2015-PR).

## Decision letter and Author response

Decision letter https://doi.org/10.7554/eLife.69058.sa1
Author response https://doi.org/10.7554/eLife.69058.sa2

## Additional files

### Supplementary files

• Transparent reporting form

### Data availability

All data generated or analysed during this study are included in the manuscript. Source data files are provided for all the figures presented in the manuscript.

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
