## [Editor Report]

This manuscript investigates the role of the repressor-element 1-silencing transcription/neuron-restrictive silencer factor (REST/NRSF) in homeostatic regulation of neuronal activity in response to chronic hyperactivity in cultured neural networks. This work demonstrates the role of REST in network firing properties and specifically the role of inhibition in this regulation. This paper is of potential interest to a broad audience of neuroscientists, interested in the mechanisms of homeostatic plasticity and in brain disorders associated with neural dyshomeostasis.

---

## [Decision Letter]

**Decision letter after peer review:**

Thank you for submitting your article "REST/NRSF drives homeostatic plasticity of inhibitory synapses in a target-dependent fashion" for consideration by *eLife*. Your article has been reviewed by 2 peer reviewers, one of whom is a member of our Board of Reviewing Editors, and the evaluation has been overseen by Lu Chen as the Senior Editor. The following individual involved in review of your submission has agreed to reveal their identity: Ana Luisa Carvalho (Reviewer #2).

Essential revisions:

In order to strengthen the conclusions of the manuscript, we recommend the authors to provide a better link between the network-wide effects of REST silencing and its synaptic effects altering excitation-inhibition balance and BDNF-dependent molecular mechanisms underlying these effects.

1. MEA experiments:

1a. Control cultures show now no stability, ~40% increase in mean firing rate was observed after 48 hr of recordings. If possible, it would be great to improve recording conditions to stabilize the control recordings (could be due to evaporation, pH changes in the recording chamber etc).

1b. Control cultures show over-compensation of mean firing rate in response to 4AP. These results are problematic since the chosen perturbation impairs mean firing rate compensation in control conditions, resulting in over-compensation. I suggest to validate this conclusion. It is possible that 4AP at the concentration used causes excitotoxity which is pathological. The authors may use different treatments to induce hyperactivity that have been previously shown in the literature to induce normal compensation of mean firing rate under control conditions (or lower concertation of 4AP). Otherwise, these results are difficult to interpret and ODN-mediated rescue of homeostatic mean firing rate compensation is relevant only for pathological, epilepsy-like conditions. It should be clearly stated.

1c. In contrast to mean firing rate, burst properties were normalized following 48hr 4AP treatment. However, ODN treatment impaired this compensation. These results suggest that REST differentially regulate mean firing rate and firing pattern. The authors should make these conclusions clearly and explain how their mechanistic data support these network-wide effects.

2. In the paper the authors use term 'homeostatic' without mentioning what network parameter is under homeostatic control. According to the data in the manuscript, only firing pattern is homeostatically preserved following 2 days of 4AP treatment. The authors did not make an attempt to link their cellular data (in the current and the previous papers) and recent network data. The discussion of this link should be added to the revised manuscript. How REST-mediated changes in excitatory and inhibitory synapses, in addition to changes in intrinsic excitability, can explain the network-wide over-compensation of mean firing rate and compensation of firing pattern following 4AP treatment.

3. The authors address possible mechanisms underlying the observed effect, and focus on BDNF and Npas4 as possible mediators. The rationale for testing a role for BDNF is that BDNF is known to strengthen inhibitory inputs, and the authors argue that it is released mainly by excitatory neurons (although conclusive evidence for this is missing). Since enhanced GABAergic transmission was only detected on inhibitory contacts to excitatory neurons, it is proposed that BDNF could be the retrograde messenger driving the specificity of the observed effect. This idea, while attractive, lacks confirmation. The issues below should be considered to strengthen the molecular part of the study:

a) Data presented in the manuscript do show that inhibition of BDNF binding to TrkB mimics the effect of REST inhibition on eIPSCs amplitude and on the PPR, as well as on the density of perisomatic inhibitory synapses onto excitatory neurons (Figures 6 and 7; but curiously not the decrease on dendritic inhibitory synapses, which is blocked when REST is inhibited). The authors additionally show that chronic treatment with 4AP to enhance activity leads to increased expression of Npas4 and BDNF, which are partially blocked by blockade of REST (Figure 8). However, an experiment showing that the effects of increased REST activity (elicited by hyperactivity) on inhibitory transmission depend on BDNF has not been performed. The evidence showing that the effects of REST on inhibitory transmission depend on BDNF needs to be provided.

b) The authors suggest that REST activation upon hyperactivity leads to Npas4 activation and consequent p1-BDNF expression. However, in Figure 8B it is shown that Npas4 is upregulated in the absence of active REST, but at a later time point (6h and onward). It is also persistently upregulated. These data do not match the protein analysis data (Figures 8I and J), in which ODN blocks the upregulation of Npas4 at 24h. The authors should explain these discrepancies.*Reviewer #1:*

The manuscript by Cosimo Prestigio et al., studies the role of the repressor-element 1-silencing transcription/neuron-restrictive silencer factor (REST/NRSF) in homeostatic regulation of neuronal activity in response to chronic hyperactivity in cultured neural networks. While previous studies by the authors established the role of REST in homeostatic regulation of intrinsic neuronal excitability and excitatory synaptic transmission, how REST affects network firing properties and the role of inhibition in this regulation has remained elusive.

The authors demonstrated that translocation of REST to the nucleus in response to hyperactivity occurs in both, excitatory and inhibitory neurons, while activating distinct transcriptional program based on the postsynaptic target. In particular, fast REST silencing by decoy oligodeoxynucleotide (ODN) specifically occluded an increase in mIPSC frequency and eIPSC properties (release probability, RRP size and amplitude) of inhibitory synapses on excitatory neurons, but not inhibitory neurons. Moreover, ODN-mediated REST silencing abolished an increase in the density of peri-somatic inhibitory synapses on excitatory neurons in response to hyperactivity. The authors demonstrated that this regulation of inhibition was mediated by NPAS4-BDNF signaling. At the network level, REST silencing differentially regulated mean firing rate and firing patterns in response to hyperactivity.

The paper is clearly written, the experimental design is elegant, combining network and single-cell measurements, and the data quality is high. This work strongly suggests that transcriptional regulation plays an important role in homeostatic regulation of network-wide firing properties. In order to strengthen the conclusions of the manuscript, I recommend the authors to provide a better link between the network-wide effects of REST silencing and its synaptic effects altering excitation-inhibition balance.*Reviewer #2:*

The manuscript by Prestigio and colleagues describes a very relevant study addressing the role of REST in homeostatic regulation of inhibitory synaptic transmission. The authors determine that in hippocampal neurons hyperactivity triggers the nuclear translocation of the transcription repressor REST, which is required for the homeostatic recovery from network hyperactivity. REST activity is also necessary for the increased GABAergic inhibition that contributes to this recovery, and which the authors show to be specific for inhibitory synapses in excitatory neurons. In a second part of the study the authors found that blocking BDNF signaling produces similar effects as blocking REST activity, and thus claim an implication of BDNF in the pathway triggered by chronic hyperactivity and leading to enhanced inhibitory activity, and which depends on REST activation.

Strengths:

This first part of the manuscript, regarding the implication of REST on homeostatic regulation of inhibitory transmission, is very convincing and well documented. The target-specific effect that was observed, with inhibitory transmission onto excitatory cells being specifically affected, is very relevant.

Weaknesses:

The implication of BDNF as a retrograde signal, originating from the postsynaptic excitatory neuron, in mediating REST effects on the homeostatic upscaling of inhibitory transmission needs to be further explored. Even though blockade of BDNF binding to TrkB replicates some of the effects of REST inhibition on preventing the effects of chronic hyperactivity on inhibitory synaptic transmission, it was not shown that the role of REST in this scenario depends on BDNF signaling.

---

## [Author Response]

Essential revisions:In order to strengthen the conclusions of the manuscript, we recommend the authors to provide a better link between the network-wide effects of REST silencing and its synaptic effects altering excitation-inhibition balance and BDNF-dependent molecular mechanisms underlying these effects.

We tried to address most of the criticisms in the revised version of the manuscript, although we realize that some questions might still have remained unanswered, due to the complexity of the epigenetic pathways underlying the hyperactivity-dependent homeostatic response. We think that the new version of the manuscript better clarifies of the REST-dependent homeostatic response. We also take the opportunity to thank the Editors and Reviewers for their constructive criticisms that we fully endorsed.

1. MEA experiments:1a. Control cultures show now no stability, ~40% increase in mean firing rate was observed after 48 hr of recordings. If possible, it would be great to improve recording conditions to stabilize the control recordings (could be due to evaporation, pH changes in the recording chamber etc).

Indeed, MEA recordings in 4AP-untreated cultures show a 40% increase of MFR, mostly due to the increased burst duration during the 48 h of observation, and this effect complicates the interpretation of the results. However, under our culture conditions, we regularly observe a time-dependent increase in activity (see, e.g., Suppl. Figure 7d in (Pozzi et al., 2013)).

Under our conditions, the healthy neural networks behave as a dynamic entity that, over the course of the in vitro maturation, increases the number and strength of its synaptic connections, as well as the intrinsic cell excitability. As a consequence, we routinely observe an increase in MFR, BF and BD, a phenomenon that is particularly strong between the second and third week in vitro.

This time-dependent increase could have been enhanced by the adoption of the 48-multiwells Axion BioSystems MEAs (1.1 x 1.1 mm recording area, instead of the wider MCS MEAs used in Pozzi et al., 2013) and by the corresponding higher cell-density (2500 cell/mm^2^) that we used. This could have contributed to the noticeable time-dependent increase of spontaneous activity, consistent with previous reports showing that the activity increase is enhanced by cell density, particularly during the third week in vitro (Biffi et al., 2013; Ito et al., 2010).

However, we agree that the mean firing rate increase of ~40% under control conditions (former Figure 2B) makes data interpretation more difficult. In the first version of the manuscript, we excluded only those wells that contained <4 active electrodes (< 5 spikes/min). We have now re-analyzed the data also excluding control wells showing a time-dependent increase of the spontaneous activity higher than 60% and 4AP-treated wells that were unresponsive to 4AP. This new analysis did not affect the final results, but significantly limited the activity increase under control conditions, also reducing overcompensation.

1b. Control cultures show over-compensation of mean firing rate in response to 4AP. These results are problematic since the chosen perturbation impairs mean firing rate compensation in control conditions, resulting in over-compensation. I suggest to validate this conclusion. It is possible that 4AP at the concentration used causes excitotoxity which is pathological. The authors may use different treatments to induce hyperactivity that have been previously shown in the literature to induce normal compensation of mean firing rate under control conditions (or lower concertation of 4AP). Otherwise, these results are difficult to interpret and ODN-mediated rescue of homeostatic mean firing rate compensation is relevant only for pathological, epilepsy-like conditions. It should be clearly stated.

As mentioned above, overcompensation was substantially reduced by the new data analysis. While the concern on 4AP excitotoxicity is reasonable, we have investigated it in detail in our previous paper (see Figure 4F of Pozzi et al., EMBO J, 2013), showing that a treatment with 100 µM 4AP for 24, 48 and 96 h does not affect cell viability.

1c. In contrast to mean firing rate, burst properties were normalized following 48hr 4AP treatment. However, ODN treatment impaired this compensation. These results suggest that REST differentially regulate mean firing rate and firing pattern. The authors should make these conclusions clearly and explain how their mechanistic data support these network-wide effects.

The chronic chemical stimulation with 4AP (100 µM; maintained for the entire period of observation, 48hr) dramatically increased network activity (Figure 2B), but also the sPSCs and firing frequency of individual neurons (see Suppl. Figure 1 of (Pecoraro-Bisogni et al., 2018)). At this low concentration (100 µM), the 4AP effect is limited to a moderate increase of AP width (Mitterdorfer and Bean, 2002) that, by activating a larger presynaptic ca^2+^ influx, enhances both EPSP and IPSP amplitudes. This could well explain why the acute effect of 4AP consists of an increase of MFR and BR, accompanied by a decrease of BD. In fact, larger EPSPs will induce more efficiently and frequently burst activity, but the IPSP rise will make feedback inhibition stronger, silencing burst activity more rapidly and generating shorter BD.

We agree that in the first version of the manuscript, the MEA results showed that REST differentially regulates MFR and firing pattern. However, our data did not show that the burst properties (BF and BD) are both normalized following 48hr 4AP treatment. In the previous version of Figure 2, only BF recovered the control condition while BD, similarly to MFR, was over-compensated. With the re-analysis of the data, the over-compensation of the MFR becomes not significant, although a moderate overcompensation of BD is preserved.

To further prove that the progressive increase of activity under control conditions and the possible over-compensation associated with the homeostatic effect is not due to experimental artifacts, see Figure 6E from Pozzi et al., 2013, where it is possible to appreciate the dramatic impairment in the build-up of the firing rate during in vitro development in cultures overexpressing REST. These data demonstrate that overexpression of exogenous REST, evoked a dramatic over-compensation of activity. Thus, it is possible that the hyperactivity-induced REST activation induces moderate over-compensation. Our current and previous data suggest that over-compensation could be a direct consequence of the multiple homeostatic processes triggered by hyperactivity that overcome the physiological activity increase due to network maturation. We better discuss these points in the paper.

2. In the paper the authors use term 'homeostatic' without mentioning what network parameter is under homeostatic control. According to the data in the manuscript, only firing pattern is homeostatically preserved following 2 days of 4AP treatment. The authors did not make an attempt to link their cellular data (in the current and the previous papers) and recent network data. The discussion of this link should be added to the revised manuscript. How REST-mediated changes in excitatory and inhibitory synapses, in addition to changes in intrinsic excitability, can explain the network-wide over-compensation of mean firing rate and compensation of firing pattern following 4AP treatment.

We realized we did not discuss how the homeostatic effects induced by hyperactivity on the inhibitory system may contribute to the previously described homeostatic effects in intrinsic excitability (Pozzi et al., 2013) and strength of excitatory transmission (Pecoraro-Bisogni et al., 2018). On a whole, 4AP-induced chronic hyperactivity induces the following compensatory responses triggered by REST, namely: (1) reduced expression of voltage-gated Na_V_1.2 channels, downregulating intrinsic excitability particularly at the level of excitatory neurons (Pozzi et al., 2013); (2) reduced strength of excitatory synapses to excitatory neurons, acting at the presynaptic level (Pecoraro-Bisogni et al., 2018); (3) increased strength and density of peri-somatic GABAergic synaptic contacts to excitatory neurons; (rts after 1hr and peaks at 4) decreased density of dendritic inhibitory synapses onto excitatory neurons (current paper).

Collectively, the first three effects highlight a dramatic alteration of the E/I balance that could explain the over-compensating homeostatic action of REST. The latter effect, instead, deserves some considerations. It has been reported that somatic IPSP is larger and has an almost mono-exponential decay while dendritic IPSP is smaller, slower, and has a pronounced second phase. (Miles et al., 1996). It is known that precisely timed somatic inhibition should limit both the number of action potentials and the time window during which firing can occur (Pouille, 2001), while dendritic inhibition mostly modulates the summation of excitatory synaptic potentials (Chiu et al., 2013).

It is possible to speculate that the enhancement of peri-somatic inhibitory inputs represents the main limitation to the repetitive discharge of APs by excitatory neurons, thus reducing MFR, BF and BD. In parallel, the decrease of dendritic inhibition, that normally controls the efficacy of afferent inputs by suppressing the generation of dendritic ca^2+^ spikes and limiting EPSC-induced depolarization (Miles et al., 1996), can favor postsynaptic integration and plasticity at dendritic level. It was previously shown that stability, as well as the compensatory feedback responses, are greater at the population compared to the single-unit level (Slomowitz et al., 2015), indicating that sim­ilar network properties may arise from multiple configurations of individual components (Marder and Goaillard, 2006; Prinz et al., 2004). Consistent with this concept, our results suggest that firing macro-stability can emerge from highly diverse combinations of synaptic strength and intrinsic neuronal properties.

3. The authors address possible mechanisms underlying the observed effect, and focus on BDNF and Npas4 as possible mediators. The rationale for testing a role for BDNF is that BDNF is known to strengthen inhibitory inputs, and the authors argue that it is released mainly by excitatory neurons (although conclusive evidence for this is missing).

We based our interpretation that glutamatergic principal neurons represent the main source of the activity-dependent release of BDNF based on previous reports (Matsumoto *et al.,* 2008; Dieni *et al.,* 2012) and on the demonstration that BDNF mRNA is unambiguously detected in cell bodies of excitatory neurons (Hofer et al., 1990; Isackson et al., 1991), while it is absent from cortical inhibitory interneurons (Cohen-Cory *et al.,* 2010; Andreska *et al.,* 2014). However, the most convincing data supporting our interpretation derive from the work of Greenberg and colleagues who clearly showed that high K^+^-induced depolarization enhanced BDNF mRNA in pure cultured excitatory neurons, while BDNF mRNA was not increased in pure cultured inhibitory neurons (Spiegel et al., 2014). Since enhanced GABAergic transmission was only detected at inhibitory contacts with excitatory neurons, it is tempting to speculate that BDNF could be the retrograde messenger driving the specificity of the observed effects. However, we agree that this is a debated point, as demonstrated, e.g., by a recent study which revealed expression of BDNF in both glutamatergic principal cells and GABAergic interneurons (Barreda Tomás et al., 2020).

Since enhanced GABAergic transmission was only detected on inhibitory contacts to excitatory neurons, it is proposed that BDNF could be the retrograde messenger driving the specificity of the observed effect. This idea, while attractive, lacks confirmation. The issues below should be considered to strengthen the molecular part of the study:a) Data presented in the manuscript do show that inhibition of BDNF binding to TrkB mimics the effect of REST inhibition on eIPSCs amplitude and on the PPR, as well as on the density of perisomatic inhibitory synapses onto excitatory neurons (Figures 6 and 7; but curiously not the decrease on dendritic inhibitory synapses, which is blocked when REST is inhibited).

The lack of effect of the inhibition of BDNF binding to TrkB on the decrease of dendritic inhibitory synapses indicates that hyperactivity triggers BDNF-dependent (enhancement of the strength and number of peri-somatic inhibitory synapses) and BDNF-independent (inhibition of the number of axo-dendritic synapses) REST-mediated responses. Although the two effects are opposite, electrophysiology only revealed the enhancement of inhibitory inputs, being mostly sensitive to synaptic currents of somatic origin. These effects are fully consistent with the data obtained by Greenberg and colleagues, who showed that high K^+^-induced hyperactivity simultaneously increased strength and number of somatic inhibitory synapses onto CA1 pyramidal neurons in a Npas4- and BNDF-dependent manner, while it decreased number and strength of dendritic inhibitory synapses in a Npas4-dependent but BDNF-independent way (Bloodgood et al., 2013). Moreover, in models of temporal lobe epilepsy, spontaneous GABAergic inhibition is increased in the soma and reduced in the dendrites of pyramidal neurons (Cossart et al., 2001). A further confirmation of the specific somatic, and not dendritic, effects of BDNF comes from the work of Erin Shuman and colleagues who found that BDNF mRNA was present in the somatic compartment of hippocampal slices or the somata of cultured rat hippocampal neurons, but it was rarely detected in the dendritic processes (Will et al., 2013).

The authors additionally show that chronic treatment with 4AP to enhance activity leads to increased expression of Npas4 and BDNF, which are partially blocked by blockade of REST (Figure 8). However, an experiment showing that the effects of increased REST activity (elicited by hyperactivity) on inhibitory transmission depend on BDNF has not been performed. The evidence showing that the effects of REST on inhibitory transmission depend on BDNF needs to be provided.

We thank the Reviewer for the suggestion. To this end, we studied the increase in peri-somatic synapses as a proxy of the homeostatic response to hyperactivity by addressing the question of whether the homeostatic effects of REST and BDNF occur in parallel (i.e., are independent mechanisms both activated by hyperactivity and converging on the same target) or they are serially connected with BDNF acting downstream of REST.

Thus, we have carried out new confocal imaging experiments where we compared the effects of the following treatments on the 4AP-induced increase of peri-somatic synapses:

1. REST inhibition

2. TrkB inhibition

3. TrkB inhibition + REST inhibition

4. REST inhibition + TrkB activation (with exogenous BDNF)

5. REST activation + TrkB activation

The results were statistically analyzed (2-way ANOVA) to demonstrate the presence of pure additive effects (in case of the “*in parallel*” model) or an occlusion (in case of the “*in series*” model). The new results that we obtained, support the hypothesis that the increase of peri-somatic synapses depends on an “*in series”* activation of REST and BDNF. In fact, combined inhibition of REST and BDNF exerts the same extent of inhibition of the homeostatic response (increase in peri-somatic synapses) of the respective single treatments. On the other hand, exogenous BDNF fully rescues the homeostatic response in the presence or absence of REST inhibition, indicating that BDNF lies downstream of REST activation.

**Author response image 1. sa2fig1:** 

b) The authors suggest that REST activation upon hyperactivity leads to Npas4 activation and consequent p1-BDNF expression. However, in Figure 8B it is shown that Npas4 is upregulated in the absence of active REST, but at a later time point (6h and onward). It is also persistently upregulated. These data do not match the protein analysis data (Figures 8I and J), in which ODN blocks the upregulation of Npas4 at 24h. The authors should explain these discrepancies.

Our data suggest that REST plays a dual role on Npas4 mRNA levels: (1) it allows an early induction of Npas4 mRNA, that starts after 1hr and peaks at 6 h; (2) it also favors the late return of Npas4 mRNA to basal levels after 24 h. Indeed, when REST is blocked by ODN, the NPAS4 mRNA increase is delayed and does not return to basal values after 24 h. We agree with the Editor that the direct comparison of Npas4 mRNA and protein levels shows some discrepancies, but the apparent mismatch is conditioned by the temporal lag between transcription and translation. This explains why Npas4 mRNA increases after 1 h of hyperactivity and is followed by a similar change in Npas4 protein after 20 h. We noted a similar time-lag for other REST target genes (vGAT and GAD67), whose mRNAs increase after 12/24 h of hyperactivity, while the protein levels change after 48 h.

**References**

Andreska T, Aufmkolk S, Sauer M, Blum R. 2014. High abundance of BDNF within glutamatergic presynapses of cultured hippocampal neurons. *Front Cell Neurosci* 0. doi:10.3389/fncel.2014.00107

Asada H, Kawamura Y, Maruyama K, Kume H, Ding RG, Kanbara N, Kuzume H, Sanbo M, Yagi T, Obata K. 1997. Cleft palate and decreased brain γ-aminobutyric acid in mice lacking the 67-kDa isoform of glutamic acid decarboxylase. *Proc Natl Acad Sci U S A* 94:6496–6499. doi:10.1073/pnas.94.12.6496

Barreda Tomás FJ, Turko P, Heilmann H, Trimbuch T, Yanagawa Y, Vida I, Münster-Wandowski A. 2020. BDNF Expression in Cortical GABAergic Interneurons. *Int J Mol Sci* 21:E1567. doi:10.3390/ijms21051567

Biffi E, Regalia G, Menegon A, Ferrigno G, Pedrocchi A. 2013. The influence of neuronal density and maturation on network activity of hippocampal cell cultures: a methodological study. *PLoS One* 8:e83899. doi:10.1371/journal.pone.0083899

Bloodgood BL, Sharma N, Browne HA, Trepman AZ, Greenberg ME. 2013. The activity-dependent transcription factor NPAS4 regulates domain-specific inhibition. *Nature* 503:121–125. doi:10.1038/nature12743

Chiu CQ, Lur G, Morse TM, Carnevale NT, Ellis-Davies GCR, Higley MJ. 2013. Compartmentalization of GABAergic inhibition by dendritic spines. *Science* 340:759–762. doi:10.1126/science.1234274

Cohen-Cory S, Kidane AH, Shirkey NJ, Marshak S. 2010. Brain-derived neurotrophic factor and the development of structural neuronal connectivity. *Developmental Neurobiology* 70:271–288. doi:10.1002/dneu.20774

Cossart R, Dinocourt C, Hirsch JC, Merchan-Perez A, De Felipe J, Ben-Ari Y, Esclapez M, Bernard C. 2001. Dendritic but not somatic GABAergic inhibition is decreased in experimental epilepsy. *Nature Neuroscience* 4:52–62. doi:10.1038/82900

Dieni S, Matsumoto T, Dekkers M, Rauskolb S, Ionescu MS, Deogracias R, Gundelfinger ED, Kojima M, Nestel S, Frotscher M, Barde Y-A. 2012. BDNF and its pro-peptide are stored in presynaptic dense core vesicles in brain neurons. *J Cell Biol* 196:775–788. doi:10.1083/jcb.201201038

Esclapez M, Houser CR. 1999. Up-regulation of GAD65 and GAD67 in remaining hippocampal GABA neurons in a model of temporal lobe epilepsy. *J Comp Neurol* 412:488–505.

Esclapez M, Tillakaratne NJ, Kaufman DL, Tobin AJ, Houser CR. 1994. Comparative localization of two forms of glutamic acid decarboxylase and their mRNAs in rat brain supports the concept of functional differences between the forms. *J Neurosci* 14:1834–1855.

Fukuda T, Aika Y, Heizmann CW, Kosaka T. 1998. GABAergic axon terminals at perisomatic and dendritic inhibitory sites show different immunoreactivities against two GAD isoforms, GAD67 and GAD65, in the mouse hippocampus: A digitized quantitative analysis. *Journal of Comparative Neurology* 395:177–194. doi:10.1002/(SICI)1096-9861(19980601)395:2<177::AID-CNE3>3.0.CO;2-#

Harney SC, Jones MV. 2002. Pre- and postsynaptic properties of somatic and dendritic inhibition in dentate gyrus. *Neuropharmacology* 43:584–594. doi:10.1016/s0028-3908(02)00169-7

Hartman KN, Pal SK, Burrone J, Murthy VN. 2006. Activity-dependent regulation of inhibitory synaptic transmission in hippocampal neurons. *Nat Neurosci* 9:642–649. doi:10.1038/nn1677

Hofer M, Pagliusi SR, Hohn A, Leibrock J, Barde YA. 1990. Regional distribution of brain-derived neurotrophic factor mRNA in the adult mouse brain. *EMBO J* 9:2459–2464.

Isackson PJ, Huntsman MM, Murray KD, Gall CM. 1991. BDNF mRNA expression is increased in adult rat forebrain after limbic seizures: temporal patterns of induction distinct from NGF. *Neuron* 6:937–948. doi:10.1016/0896-6273(91)90234-q

Ito D, Tamate H, Nagayama M, Uchida T, Kudoh SN, Gohara K. 2010. Minimum neuron density for synchronized bursts in a rat cortical culture on multi-electrode arrays. *Neuroscience* 171:50–61. doi:10.1016/j.neuroscience.2010.08.038

Kash SF, Johnson RS, Tecott LH, Noebels JL, Mayfield RD, Hanahan D, Baekkeskov S. 1997. Epilepsy in mice deficient in the 65-kDa isoform of glutamic acid decarboxylase. *Proc Natl Acad Sci USA* 94:14060–14065. doi:10.1073/pnas.94.25.14060

Lau CG, Murthy VN. 2012. Activity-dependent regulation of inhibition via GAD67. *J Neurosci* 32:8521–8531. doi:10.1523/JNEUROSCI.1245-12.2012

Marder E, Goaillard J-M. 2006. Variability, compensation and homeostasis in neuron and network function. *Nat Rev Neurosci* 7:563–574. doi:10.1038/nrn1949

Matsumoto T, Rauskolb S, Polack M, Klose J, Kolbeck R, Korte M, Barde Y-A. 2008. Biosynthesis and processing of endogenous BDNF: CNS neurons store and secrete BDNF, not pro-BDNF. *Nat Neurosci* 11:131–133. doi:10.1038/nn2038

Miles R, Tóth K, Gulyás AI, Hájos N, Freund TF. 1996. Differences between somatic and dendritic inhibition in the hippocampus. *Neuron* 16:815–823. doi:10.1016/s0896-6273(00)80101-4

Mitterdorfer J, Bean BP. 2002. Potassium currents during the action potential of hippocampal CA3 neurons. *J Neurosci* 22:10106–10115.

Pecoraro-Bisogni F, Lignani G, Contestabile A, Castroflorio E, Pozzi D, Rocchi A, Prestigio C, Orlando M, Valente P, Massacesi M, Benfenati F, Baldelli P. 2018. REST-Dependent Presynaptic Homeostasis Induced by Chronic Neuronal Hyperactivity. *Mol Neurobiol* 55:4959–4972. doi:10.1007/s12035-017-0698-9

Pouille F. 2001. Enforcement of Temporal Fidelity in Pyramidal Cells by Somatic Feed-Forward Inhibition. *Science* 293:1159–1163. doi:10.1126/science.1060342

Pozzi D, Lignani G, Ferrea E, Contestabile A, Paonessa F, D’Alessandro R, Lippiello P, Boido D, Fassio A, Meldolesi J, Valtorta F, Benfenati F, Baldelli P. 2013. REST/NRSF-mediated intrinsic homeostasis protects neuronal networks from hyperexcitability. *EMBO J* 32:2994–3007. doi:10.1038/emboj.2013.231

Prinz AA, Bucher D, Marder E. 2004. Similar network activity from disparate circuit parameters. *Nat Neurosci* 7:1345–1352. doi:10.1038/nn1352

Ramírez M, Gutiérrez R. 2001. Activity-dependent expression of GAD67 in the granule cells of the rat hippocampus. *Brain Res* 917:139–146. doi:10.1016/s0006-8993(01)02794-9

Rutherford LC, DeWan A, Lauer HM, Turrigiano GG. 1997. Brain-derived neurotrophic factor mediates the activity-dependent regulation of inhibition in neocortical cultures. *J Neurosci* 17:4527–4535.

Sexton CA, Penzinger R, Mortensen M, Bright DP, Smart TG. 2021. Structural determinants and regulation of spontaneous activity in GABAA receptors. *Nat Commun* 12:5457. doi:10.1038/s41467-021-25633-0

Slomowitz E, Styr B, Vertkin I, Milshtein-Parush H, Nelken I, Slutsky M, Slutsky I. 2015. Interplay between population firing stability and single neuron dynamics in hippocampal networks. *eLife* 4. doi:10.7554/*eLife*.04378

Spiegel I, Mardinly AR, Gabel HW, Bazinet JE, Couch CH, Tzeng CP, Harmin DA, Greenberg ME. 2014. Npas4 regulates excitatory-inhibitory balance within neural circuits through cell-type-specific gene programs. *Cell* 157:1216–1229. doi:10.1016/j.cell.2014.03.058

Stork O, Ji FY, Kaneko K, Stork S, Yoshinobu Y, Moriya T, Shibata S, Obata K. 2000. Postnatal development of a GABA deficit and disturbance of neural functions in mice lacking GAD65. *Brain Res* 865:45–58. doi:10.1016/s0006-8993(00)02206-x

Swanwick CC, Murthy NR, Kapur J. 2006. Activity-dependent scaling of GABAergic synapse strength is regulated by brain-derived neurotrophic factor. *Mol Cell Neurosci* 31:481–492. doi:10.1016/j.mcn.2005.11.002

Tian N, Petersen C, Kash S, Baekkeskov S, Copenhagen D, Nicoll R. 1999. The role of the synthetic enzyme GAD65 in the control of neuronal γ-aminobutyric acid release. *Proc Natl Acad Sci U S A* 96:12911–12916. doi:10.1073/pnas.96.22.12911

Will TJ, Tushev G, Kochen L, Nassim-Assir B, Cajigas IJ, tom Dieck S, Schuman EM. 2013. Deep Sequencing and High-Resolution Imaging Reveal Compartment-Specific Localization of Bdnf mRNA in Hippocampal Neurons. *Sci Signal* 6:rs16.